# Design of 1 × 2 MIMO Palm Tree Coplanar Vivaldi Antenna in the E-Plane with Different Patch Structure

Nurhayati Nurhayati [1,*], Eko Setijadi [2], Alexandre Maniçoba de Oliveira [3], Dayat Kurniawan [4] and Mohd Najib Mohd Yasin [5]

1   Department of Electrical Engineering, Universitas Negeri Surabaya, Surabaya 60231, Indonesia
2   Departement of Electrical Engineering, Institut Tenologi Sepuluh Nopember, Surabaya 60111, Indonesia
3   Laboratório James Clerk Maxwell de Micro-ondas e Eletromagnetismo Aplicado (LABMAX),
    Instituto Federal de Educacao, Ciencia e Tecnologia de Sao Paulo, Cubatão 11533-160, Brazil
4   Research Center for Telecommunication, National Research and Innovation Agency (BRIN),
    Bandung 40135, Indonesia
5   Advanced Communication Engineering (ACE) Centre of Excellence, Faculty Electronic Engineering
    Technology, Universiti Malaysia Perlis (UniMAP), Kangar 01000, Perlis, Malaysia
*   Correspondence: nurhayati@unesa.ac.id

**Abstract:** In this paper, 1 × 2 MIMO of Palm Tree Coplanar Vivaldi Antenna is presented that simulated at 0.5–4.5 GHz. Some GPR applications require wideband antennas starting from a frequency below 1 GHz to overcome high material loss and achieve deeper penetration. However, to boost the gain, antennas are set up in MIMO and this is costly due to the large size of the antenna. When configuring MIMO antenna in the E-plane, there is occasionally uncertainty over which antenna model may provide the optimum performance in terms of return loss, mutual coupling, directivity, beam squint, beam width, and surface current using a given substrate size. However, the configuration of E-plane antenna in MIMO has an issue of mutual coupling if the distance between elements is less than 0.5λ. Furthermore, it produces grating lobes at high frequencies. We implement several types of patch structures by incorporating the truncated, tilt shape, Hlbert and Koch Fractal, Exponential slot, Wave slot, the lens with elips, and metamaterial slot to the radiator by keeping the width of the substrate and the shape of the feeder. The return loss, mutual coupling, directivity, beam squint, beamwidth, and surface current of the antenna are compared for 1 × 2 MIMO CVA. A continuous patch MIMO has a spacing of 0.458λ at 0.5 GHz, which is equivalent to its element width. From the simulation, we found that Back Cut Palm Tree (BCPT) and Horizontale Wave Structure Palm Tree (HWSPT) got the best performance of return loss and mutual scattering at low-end frequency respectively. The improvement of directivity got for Metamaterial Lens Palm Tree (MLPT) of 4.453 dBi if compared with Regular Palm Tree-Coplanar Vivaldi Antena (RPT) at 4 GHz. Elips Lens Palm Tree (ELPT) has the best beam squint performance across all frequencies of 0°. It also gots the best beamwidth at 4.5 GHz of 3.320. In addition, we incorporate the MLPT into the radar application.

**Keywords:** bandwith; Coplanar Vivaldi Antenna; mutual coupling; radiation pattern

## 1. Introduction

The antenna is an important component of the telecommunications system because it is the final part of the process of sending and receiving data. Presently, there is a continuous increase in data transmission hence research for improving antenna performance continues to rise. Many antenna studies have been carried out for microwave imaging [1–3] military or electronic warfare [4,5], Wifi [6,7], 5G telecommunications [8–11], vehicle communication [12–14], Maritim, airborne [15,16], and RADAR applications [17–19]. The studies conducted on RADAR applications including GPR applications usually work in the low frequency between 500 MHz to 3 GHz [20,21]. Antenna research that discusses its usage in

an extensive frequency has been conducted [22–24], however, research on planar antennas that perform at low frequencies below 1 GHz is still quite restricted.

An antenna that has higher bandwidth would contribute immensely to the high imaging resolution of the radar system. UWB imaging system is designed to detect and see the object of interest in a structure that can be classified as through-the-wall imaging biomedical imaging and ground-penetrating radar. The higher the bandwidth the better the image resolution. The lower the frequency the larger the wavelength, so a radar with a very large wavelength will not be able to locate anything very precisely. The lower frequency the poorer the resolution. On the other hand, the lower frequency has its advantage, particularly when using the radar for GPR or through-the-wall imaging applications as this would ease the high material losses that will be encountered when using a higher frequency range and also guarantee deeper penetration of the electromagnetic wave. Furthermore, radar and telecommunications applications require antennas that work with wide bandwidth and high gain.

There have also been studies on how to boost its bandwidth and gain by changing its feeding, ground plane, and radiator, including the inclusion of corrugated slots. Vivaldi is one of the planar antennas that can work in a wide bandwidth. This antenna produces high gain coverage by providing corrugated [25], metamaterial [26,27], lens [28,29], fractal structures [30], and others. In addition to modifying the antenna elements, the increasing gain can be done by arranging the antenna in the form of an array and MIMO [31] as discussed in [32–36]. MIMO (Multiple Input Multiple Output) uses multiple antennas in the transmitter and receiver to improve its gain and data output. The MIMO antenna can be applied for radio astronomy applications, GPR, through-wall imaging applications [37], microwave tomography [38], and 5G applications [39]. The antenna arrangement in the form of MIMO on wideband antennas must pay attention to the mutual coupling between neighboring components in the low-end frequency because it affects the scattering characteristics.

Research on the method of reducing mutual antenna coupling, especially for Vivaldi, has been discussed by [40] using the corrugated slot technique, in [41] by multiple notch structure in the ground plane, and [42] by a triangular director. Mutual coupling is a significant problem in the E-plane because surface current flows in the neighboring element by the continuous patch, causing coupling between elements, but in the H-plane, the coupling is caused by an electric field flowing through the air. Mutual coupling antenna with very wide bandwidth will be risky at low-end frequencies while for high frequencies there is usually no problem. This is at low frequencies, the antenna has a longer wavelength. Therefore, to reduce mutual coupling between elements, the antenna must be more than half the wavelength long. If the antenna is arranged in MIMO for wideband antennas, there will be a trade-off between the mutual coupling performance and the radiation pattern performance at low and high frequencies. At low frequencies, The Mutual coupling performance of the antenna is good if it has the mutual scattering parameter $S_{21} < -20$ dB or the isolation of more than 20 dB. When the distance between elements is too large relative to the wavelength at high frequencies, a free lobe will emerge, thereby deteriorating the performance of the radiation pattern. Many different types of antennas may be used in radar and communications applications, including patch antennas [3], monopole antennas [6,8], and 3D antennas [5]. However, some of those antennas have an omnidirectional radiation pattern, or if they do have a directional radiation pattern, the directivity is minimal. Vivaldi antenna has advantages such as a planar antenna, wide bandwidth, and directional radiation pattern. There are several types of Vivaldi antennas including Coplanar Vivaldi Antenna [43,44], Antipodal Vivaldi Antenna (AVA) [45], and Balanced Antipodal Vivaldi Antenna (BAVA) [46]. The AVA is frequently mentioned in contrast to the CVA, even though the CVA also offers benefits in its gain performance. Additionally, the Vivaldi antenna's elements are explained in greater detail than those of the MIMO antenna. As a consequence, a discussion about the performance of the CVA antenna in MIMO is also suggested.

According to the preceding explanation, many radar applications require antennas that can operate at low frequencies, and the size of antennas that operate at low frequencies is often large. Meanwhile, the production of huge antennas and MIMO setups is prohibitively expensive. Aside from that, other researchers have done studies on Vivaldi antennas with several models, but very few have compared multiple models with the same substrate size. And antenna designers are occasionally perplexed about which antenna model to utilize to optimize certain performances. As a result, a discussion about the performance of the Vivaldi antenna is required, including return loss, mutual coupling, beam squint, and a beamwidth of the antenna of the same size, so that the antenna performance results can be used as a recommendation for which antenna design accommodates the required performance.

As a result, our contribution is to create 15 different slot structures to the Palm Tree CVA to investigate the effect of slot shape on mutual coupling reduction as well as the effect of different slot structure shapes on the performance of the radiation pattern using the same substrate width. We only discuss mutual coupling in the E-field in this discussion because mutual coupling in the E-field is a major problem, particularly at low frequencies. After all, surface currents can flow directly to adjacent elements, whereas mutual coupling in the H field can occur due to electric field induction through the air. We discovered the position and shape of the truncated slot that provides the best performance in terms of return loss performance, the wavy slot shape that provides the best performance in terms of mutual coupling at low frequencies, and the size of the elliptical structure that can provide 0° beam squint performance at all entire frequencies (0.5–4.5 GHz) while also providing the best beamwidth. Furthermore, we developed a metamaterial in the form of a split ring resonator that can boost gain by 4.451 dBi when compared to the RPT-CVA without giving a slot structure, and we evaluate it with measurement data, apply it to radar, and compare it to relevant research.

In this study, we compared the performance of the return loss, mutual coupling, side lobe level, beam squint, and a beamwidth of 15 types of $1 \times 2$ MIMO Palm Tree Coplanar Vivaldi Antenna using the same feeding shape and the substrate width in the E-plane. The antennas compared are Regular Palm Tree-Coplanar Vivaldi Antena (RPT-CVA), Front Cut Palm Tree (FCPT-CVA), Middle Cut Palm Tree (MCPT-CVA), Back Cut Palm Tree (BCPT-CVA), Complete Cut Palm Tree (CCPT-CVA), Left Tilt Palm Tree (LTPT-CVA), Right Tilt Palm Tree (RTPT-CVA), Hilbert Fractal Structure Palm Tree (HFSPT-CVA), Koch Fractal Structure Palm Tree (KFSPT-CVA), Exponential Slot Edge Palm Tree (ESEPT-CVA), Vertical Wave Structure Palm Tree (VWPT-CVA), Horizontale Wave Structure Palm Tree (HWPT-CVA), Regular Lens Palm Tree (RLPT-CVA), Elips Lens Palm Tree (ELPT-CVA) and Metamaterial Lens Palm Tree (MLPT-CVA).

This paper is organized as follows: Section 2 discusses antenna design, Section 3 discussesResults and Discussion, Section 4 discusses Measurement and comparison of related antenna, and Section 5 conclusion.

## 2. Antenna Configuration

This study designed $1 \times 2$ antenna Palm Tree Coplanar Vivaldi antenna in the E-plane, as shown in Figures 1 and 2, using the same type and size of the substrate. The antenna is designed with an FR-4 substrate with a thickness of 1.6 mm, while the radiator and feeding shape is made of copper with a thickness of 0.035 mm. The dimensions of the patch element are 250 mm $\times$ 250 mm $\times$ 0.035 mm but it has an additional substrate width of 12.5 mm on both sides of the antenna width to reduce the mutual coupling. The antenna was simulated in the frequency range between 500 MHz–4.5 GHz with the dimension as illustrated in Table 1. Equation (1) is used to determine the tapered side on the bottom of PT CVA, the flared region, and the ESE structure. The $C_1$ and $C_2$ are constants, and $R$ is the exponential growth rate with the value of $R_1$, $R_2$, $R_3$, and $R_4$ shown in Table 1 (based on Figure 1a. The beginning and ending points of an exponential curve are $x_1$, $y_1$, $x_2$, and $y_2$ [44].

$$y = C_1 e^{Rx} + C_2, \quad C_1 = \frac{y_2 - y_1}{e^{Rx_2} - e^{Rx_1}}, \quad C_2 = \frac{y_1 e^{Rx_2} - y_2 e^{Rx_1}}{e^{Rx_2} - e^{Rx_1}} \tag{1}$$

Figure 1h delivers the Hilbert curve structure in the third iteration with the total length of the slot ($l_h$), the line segment ($dd$), and iteration ($in$) follows Equation (2) [47]. In this case, we use 3rd iteration.

$$dd = \frac{l_h}{2^{in} - 1} \tag{2}$$

Figure 2c,d show the vertical and horizontal wave slots. The Constant of the wavy slot in Equation (3) is $B_1 = 2$, $B_2 = 1$, $B_3 = 1$, $B_4 = 5$, and $B_5 = 36$. The depth of the wave, the number of waves, and the length of the wave slot can all be modified by changing the value of $B_n$ [48].

$$A(t) = B_1 \left( B_2 + B_3 \ \cos \left( \frac{B_4 \pi t}{B_5} \right) \right) \tag{3}$$

**Table 1.** Parameter dimension of the antenna.

| Dimension (mm) | | | | | | | | | |
|---|---|---|---|---|---|---|---|---|---|
| Par | Dim | Par | Dim | Par | Dim | Par | Dim | Par | Dim |
| a | 550 | k | 100 | u | 25 | E | 10 | O | 12 |
| b | 250 | l | 50 | v | 93.53 | F | 188 | P | 120° |
| c | 135 | m | 99.78 | w | 6 | G | 30 | Q | 12 |
| d | 55 | n | 38 | x | 17.9 | H | 150 | R | 10 |
| e | 0.5 | o | 30° | y | 162 | I | 50 | S | 16 |
| f | 90 | p | 30° | z | 25.11 | J | 51 | $R_1$ | 0.03 |
| g | 14 | q | 35 | A | 25 | K | 1 | $R_2$ | 0.05 |
| h | 1005 | r | 25 | B | 62.39 | L | 6 | $R_3$ | −0.2 |
| i | 50 | s | 41 | C | 5 | M | 0.7 | $R_4$ | −02 |
| j | 25 | t | 5 | D | 25 | N | 0.4 | | |

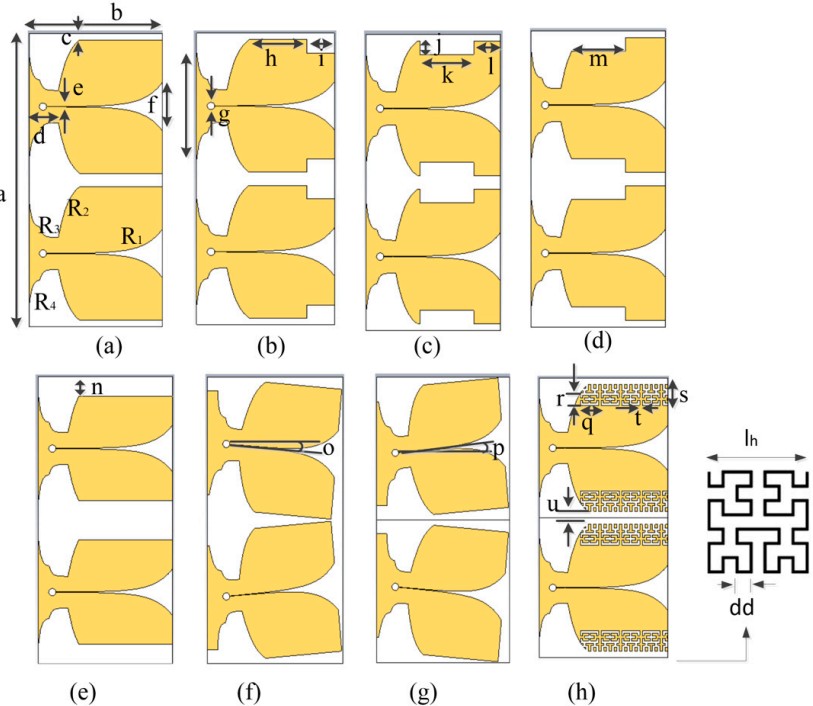

**Figure 1.** The 1 × 2 Coplanar Vivaldi Antenna of (**a**) Regular Palm Tree (RPT-CVA), (**b**) Front Cut Palm Tree (FCPT-CVA), (**c**) Middle Cut Palm Tree (MCPT-CVA), (**d**) Back Cut Palm Tree (BCPT-CVA), (**e**) Complete Cut Palm Tree (CCPT-CVA), (**f**) Left Tilt Palm Tree (LTPT-CVA), (**g**) Right Tilt Palm Tree (RTPT-CVA), (**h**) Hilbert Fractal Structure Palm Tree (HFSPT-CVA).

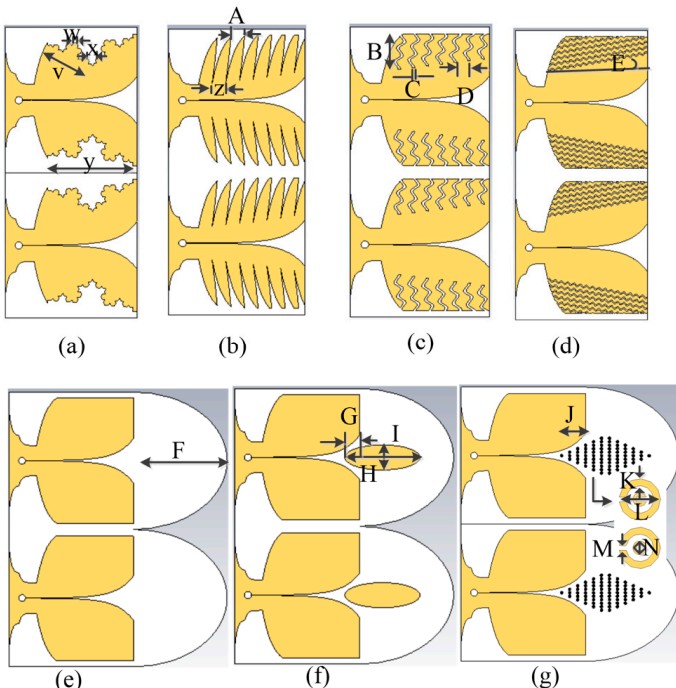

**Figure 2.** The $1 \times 2$ MIMO Coplanar Vivaldi Antenna of (**a**) Koch Fractal Structure Palm Tree (KFSPT-CVA), (**b**) Exponential Slot Edge Palm Tree (ESEPT-CVA), (**c**) Vertical Wave Structure Palm Tree (VWPT-CVA), (**d**) Horizontale Wave Structure Palm Tree (HWPT-CVA), (**e**) Regular Lens Palm Tree (RLPT-CVA), (**f**) Elips Lens Palm Tree (ELPT-CVA), (**g**) Metamaterial Lens Palm Tree (MLPT-CVA).

## 3. Results and Discussion

### 3.1. Scattering Parameter Performance

The $S_{11}$ and $S_{21}$ performance of RPT, FCPT, and MCPT-CVA are displayed in Figure 3. The best $S_{11}$ is obtained from MCPT, at the low-end frequency. According to the simulation results displayed in Figure 3a, the low-end frequencies for RPT, FCPT, and MCPT-CVA that got $S_{11}$ −10 dB are 0.52 GHz, 0.53 GHz, and 0.5 GHz. FCPT experienced poor return loss due to $S_{11}$ exceeding −10 dB for frequencies 0.62 GHz to 0.81 GHz. The right side of Figure 1 shows the $S_{21}$ performance of the RPT, FCPT, and MCPT. As can be observed, in the first low-end frequency at −20 dB the mutual scattering parameters for RPT, FCPT, and MCPT are 0.769 GHz, 0.848 GHz, and 0.58 GHz respectively. The best $S_{21}$ is achieved for MCPT at low-end frequencies. However, the best $S_{21}$ in overall frequency is acquired for FCPT as shown in Figure 3. The mutual scattering parameter has a lesser value at higher frequencies. In this study, an antenna with a substrate element's width of 275 mm was created, with frequencies of 0.5 GHz and 4.5 GHz and equal to the wavelengths of 600 mm and 66.67 mm, respectively. If the antenna has spacing between elements 275 mm so electrically, the antenna has a size of 0.458λ at a frequency of 0.5 GHz and 4.125λ at a frequency of 4.5 GHz. The mutual coupling of the antenna will be large if it has a size of fewer than 0.5λ (at the low-end frequency). Wideband and Ultra Wideband and also Super wideband antennas will experience mutual coupling issues at low frequencies. The greater the frequency, the less mutual coupling. However, it yields the problem of the grating lobe at high frequency. The performance of return loss and mutual scattering characteristics to the RPPT, BCPT, and CCPT in $1 \times 2$ MIMO antenna are assessed in Figure 3b. The best return loss performance is attained for BCPT. At 0.5 GHz, the antenna's $S_{11}$ is −10.736 dB. At a frequency of 0.79 GHz, it has an $S_{11}$ of −51.094 dB. The worst return loss for CCPT appears at low frequency. It demonstrates that at 0.699 GHz, it exhibits $S_{11}$ −5.985 dB.

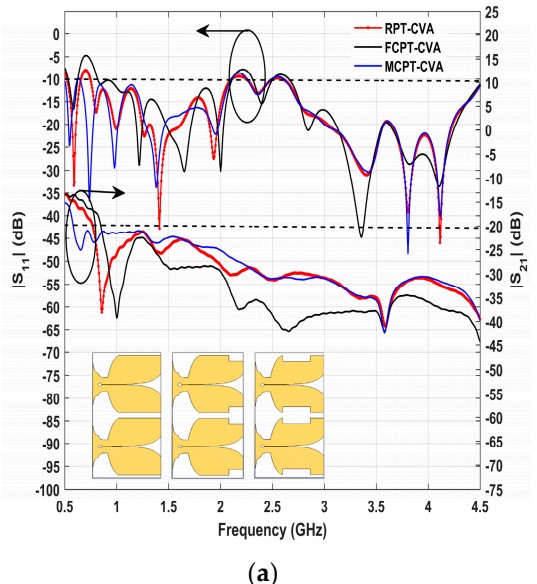

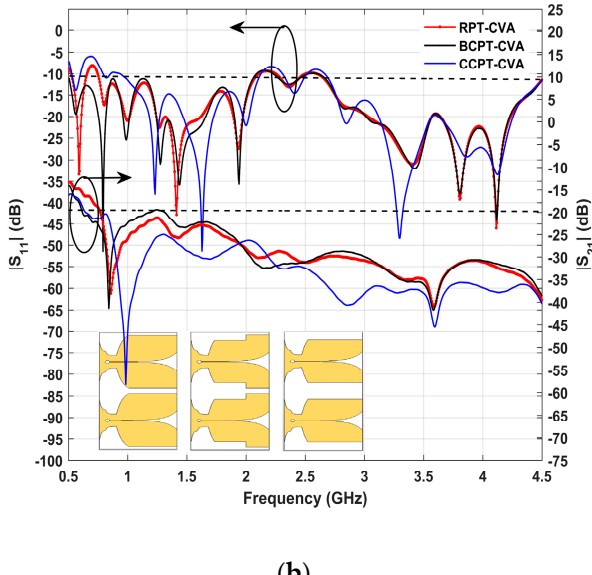

(**a**)                                                                 (**b**)

**Figure 3.** $S_{11}$ and $S_{21}$ performance of $1 \times 2$ MIMO (**a**). Regular Palm Tree Coplanar Vivaldi Antena (RPT-CVA), Front Cut Palm Tree (FCPT-CVA), Middle Cut Palm Tree (MCPT-CVA), and (**b**). $S_{11}$ and $S_{21}$ of Regular Palm Tree Coplanar Vivaldi Antena (RPT-CVA), Back Cut Palm Tree (BCPT-CVA), Complete Cut Palm Tree (CCPT-CVA).

However, CCPT has an $S_{11}$ of $-51.359$ dB at 1.628 GHz. The $S_{21}$ of RPT, BCPT, and CCPT might be visible on Figure 3b's bottom side. The best $S_{21}$ performance is in BCPT, CCPT, and RPT CVA. Meanwhile, the CCPT CVA received $S_{21}$ at 0.989 GHz of $-55$ dB. The CCPT performs well in $S_{21}$ because it has the largest truncated area without copper on both edges of the patch.

Figure 4a shows the comparison of return loss performance and mutual scattering performance of RPT, LTPT, and RTPT CVA. From Figure 4a, it could be seen that LTPT has the best performance, due to covering $S_{11}$ below $-10$ dB almost in the all-frequency range from 0.5–4.5 GHz. However, the bottom of Figure 4a shows that LTPT has the worst $S_{21}$ performance. At 0.5 GHz LTPT has $S_{21}$ $-11.491$ dB. LTPT has a distance between the feeding point is $0.495\lambda$ and RTPT of $0.422\lambda$ The distance between the two feeds for LTPT is getting farther away from the distance between the two centers of the tapered slot. RTPT has the best $S_{21}$ at 0.885 GHz of $-47.506$. The electric field between two tapered slots will propagate away for RTPT while for LTPT. Figure 4b shows that the HFSPT and KFSPT have better performance of return loss than RPT at low-end frequency. At 0.5 GHz the $S_{11}$ performance of RPT, HFSPT, and KFSPT are $-8.311$ dB, $-12.442$ dB, and $-14.33$ dB respectively. At 4.1 GHz KFSPT result $S_{11}$ of $-51.093$ dB. $S_{21}$ performance of RPT, HFSPT, and KFSPT is declared at the bottom of Figure 4b. At 0.5 GHz the $S_{21}$ of RPT, HFSPT, and KFSPT are $-8.311$ dB, $-14.502$ dB, and $-15.291$ dB. It can be concluded that the KFSPT has the best $S_{21}$ performance at low-end frequency.

The performance of $S_{11}$ and $S_{21}$ for RPT, ESEPT, VWSPT, and HWSPT is shown in Figure 5a. We can see that the best $S_{11}$ performance is found for RPT. However, ESEPT, VWSPT, and HWSPT have $S_{11}$ of more than $-10$ dB in some low frequencies. At 0.5 GHz ESEPT has $S_{11}$ $-16.859$ dB but at 0.517 GHz to 0.628 GHz. it has an $S_{11}$ of more than $-10$ dB as well as the structure of VWSPT and HWSPT.

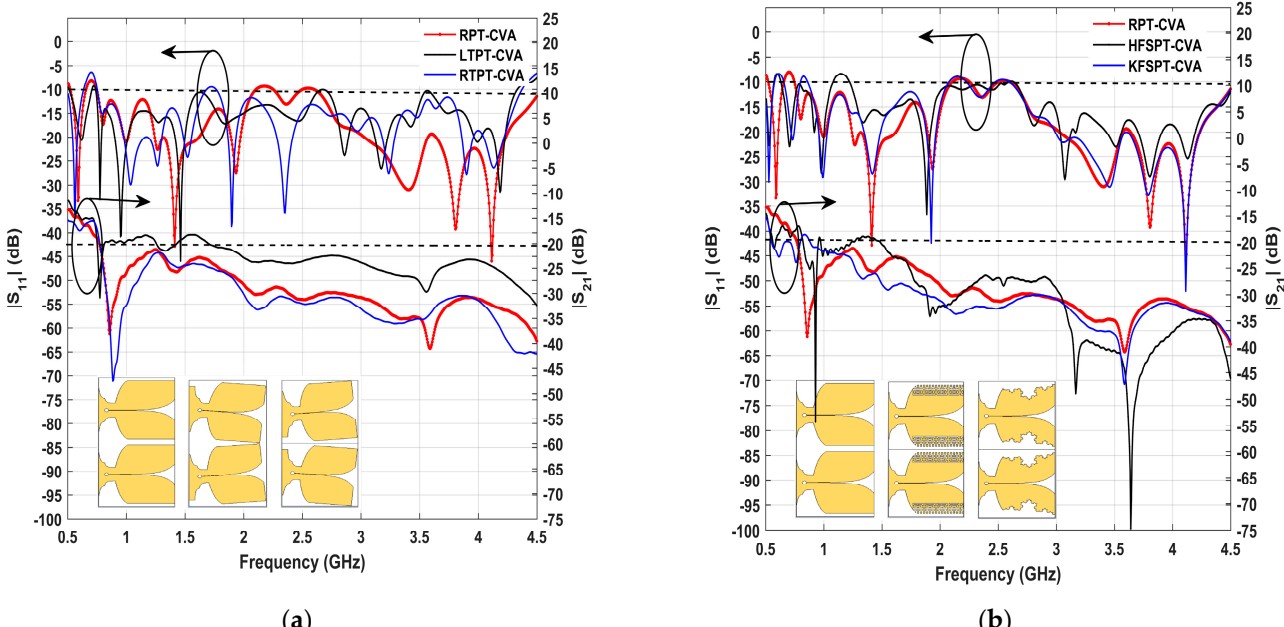

**(a)**            **(b)**

**Figure 4.** $S_{11}$ and $S_{21}$ performance of $1 \times 2$ MIMO (**a**). Regular Palm Tree-Coplanar Vivaldi Antena (RPT-CVA), Left Tilt Palm Tree (LTPT-CVA), Right Tilt Palm Tree (RTPT-CVA) and (**b**). Regular Palm Tree-Coplanar Vivaldi Antena (RPT-CVA), Hilbert Fractal Structure Palm Tree (HFSPT-CVA), Koch Fractal Structure Palm Tree (KFSPT-CVA).

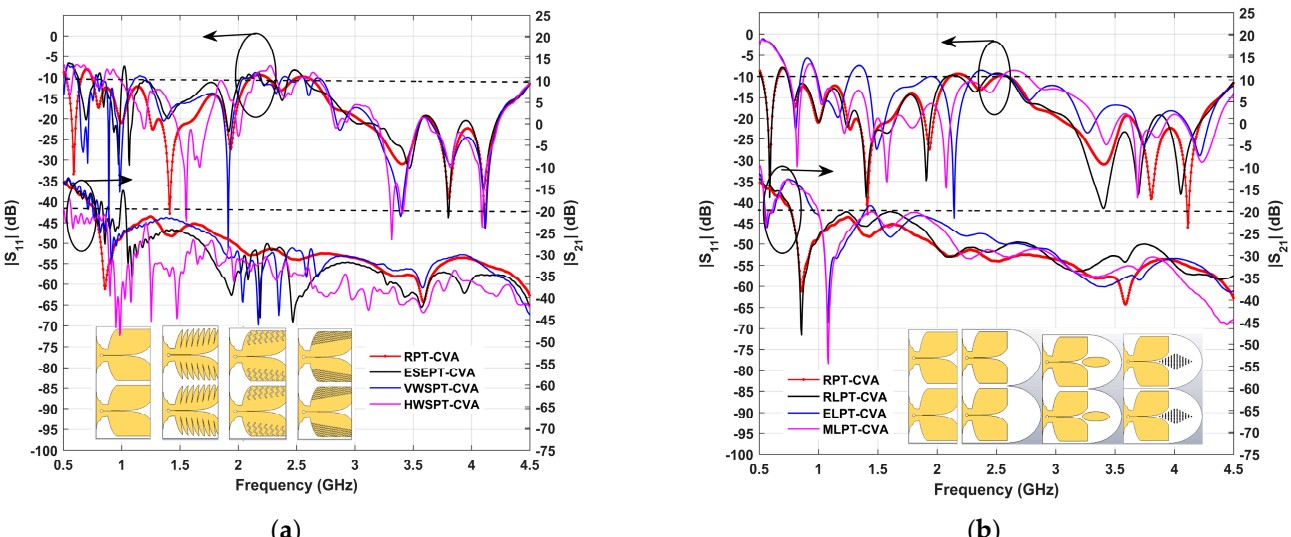

**(a)**            **(b)**

**Figure 5.** $S_{11}$ and $S_{21}$ performance of $1 \times 2$ MIMO (**a**). Regular Palm Tree-Coplanar Vivaldi Antena (RPT-CVA), Exponential Slot Edge Palm Tree (ESEPT-CVA), Vertical Wave Structure Palm Tree (VWPT-CVA), and Horizontale Wave Structure Palm Tree (HWPTCVA) and (**b**) $S_{11}$ and $S_{21}$ of Regular Palm Tree-Coplanar Vivaldi Antena (RPT-CVA), Elips Lens Palm Tree (ELPT-CVA), and Metamaterial Lens Palm Tree (MLPT-CVA).

The return loss performance of the low frequency will be impacted by the corrugated structure or wave structure. The electric field will be trapped between the corrugated and wave structure. However, the Mutual scattering of HWSPT got the best performance than others. The $S_{21}$ of RPT, ESEPT, and VWSPT is almost the same. Adding corrugated and wave structures in the vertical direction will affect the $S_{21}$ performance at low frequencies. The return loss performance of RPT and RLPT is almost the same, likewise the performance of ELPT and MLPT-CVA. It could be shown in Figure 5b that by adding the Ellips and

metamaterial structure in the mouth flared of the two tapered slots, the return loss performance in the low-end frequency becomes worst in the low-end frequency than without adding structure. From Figure 5b, it can also be seen that RPT and RLPT have almost the same return loss at frequencies below 3 GHz, but above 3 GHz there are differences. $S_{21}$ performance of RPT, RLPT, ELPT, and MLPT can be seen at the bottom of Figure 5b. Even though ELPT and MLPT appear to have $S_{21}$ below $-20$ dB at low-end frequency but only a few frequencies are covered. All four antennas have poor mutual catering parameters at low frequencies, even if at 0.895 GHz MLPT has $S_{21}$ of $-54.432$ dB. According to the overall scattering parameter simulation findings for the 15 antenna types, BCPT has the best $S_{11}$ performance at low frequencies, while HWSPT has the most effective $S_{21}$ performance at low-end frequencies. Adding the structure in the patch will affect the electric field so that it interferes with the scattering parameters.

### 3.2. Radiation Pattern Performance

### 3.2.1. Directivity Performance

The directivity of the element and $1 \times 2$ RPT, $1 \times 2$ of FCPT, MCPT, BCPT, and CCPT is displayed in Figure 6a. At 2 GHz, by arranging the antenna into a MIMO, there is an improvement in RPT directivity of 3.2 dBi. At 2 GHz, the directivity of $1 \times 2$ RPT and $1 \times 2$ FCPT is 8.894 dBi and 11.587 dBi. This means there is an improvement in a gain of 2.693 dBi. However, at a frequency above 2 GHz, the gain decreases. The highest directivity is obtained for $1 \times 2$ BCPT at 4 GHz of 12 dBi.

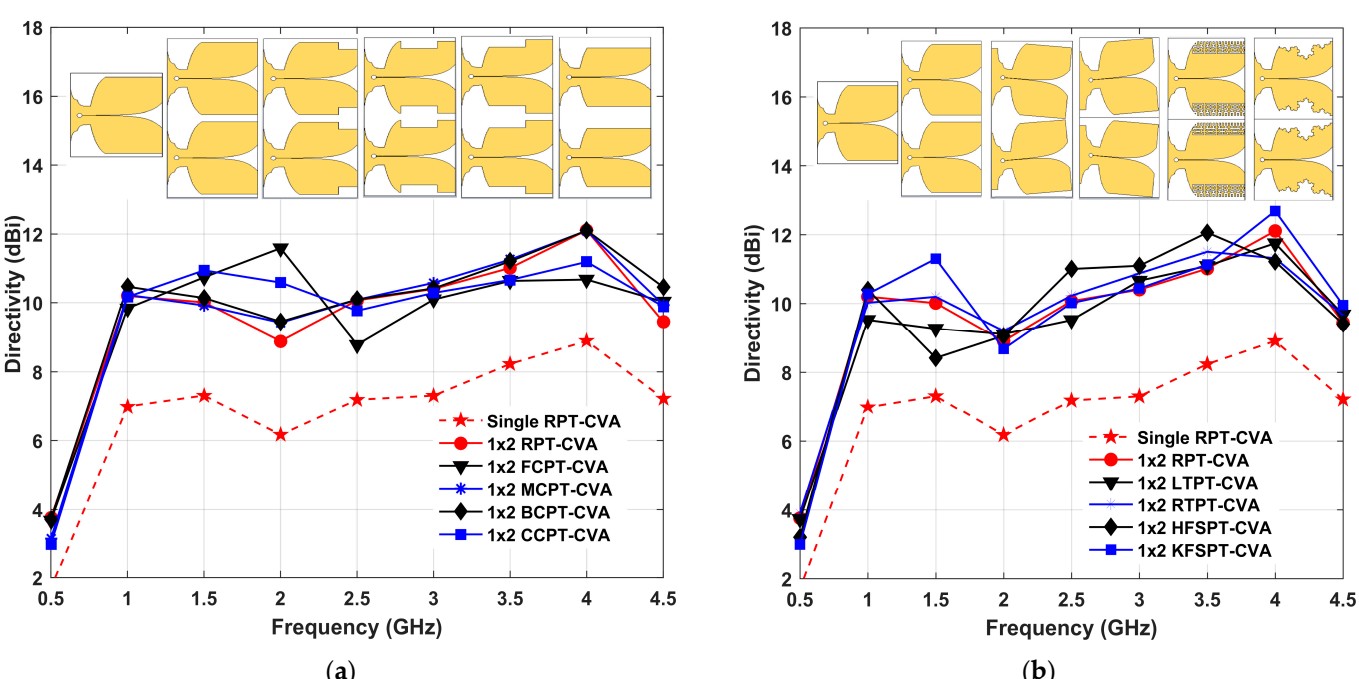

**Figure 6.** Directivity of: (**a**). Single and $1 \times 2$ Regular Palm Tree (RPT-CVA), $1 \times 2$ Front Cut Palm Tree (FCPT-CVA), $1 \times 2$ Middle Cut Palm Tree (MCPT-CVA), $1 \times 2$ Back Cut Palm Tree (BCPT-CVA), $1 \times 2$ Complete Cut Palm Tree (CCPT-CVA) and (**b**). Single and $1 \times 2$ of Regular Palm Tree (RPT-CVA), $1 \times 2$ Left Tilt Palm Tree (LTPT-CVA), $1 \times 2$ Right Tilt Palm Tree (RTPT-CVA), $1 \times 2$ Hilbert Fractal Structure Palm Tree (HFSPT-CVA), $1 \times 2$ Koch Fractal Structure Palm Tree (KFSPT-CVA).

Figure 6b shows the directivity comparison between single and $1 \times 2$ RPT, $1 \times 2$ of LTPT, RTPT, HFSPT, and KFSPT. It demonstrates that the directivity changes slightly by tilting the antenna. It means there is an improvement in directivity by making the antenna position slightly tilted left and right, in this case, tilted 5 degrees, then the directivity does not change much. But at a frequency below 3.5 GHz, the $1 \times 2$ RTPT has a better performance of directivity than $1 \times 2$ LTPT and $1 \times 2$ RPT. At low frequencies, it can be a

consideration for arranging the antenna with a larger tilt angle outward position so that the electric field coming out of the two tapered slots does not affect each other. Figure 6b shows the directivity of 1 × 2 KFSPT of 11.299 dBi while 1 × 2 RPT of 8.894 dBi. At Frequency 4 GHz the 1 × 2 KFSPT has the best directivity of 12.108 dBi.

The directivity performance of single and 1 × 2 RPT, 1 × 2 of ESEPT, VWSPT, and HWSPT can be observed a Figure 7a. 1 × 2 ESEPT has better performance of directivity than 1 × 2 RPT and 1 × 2 VWSPT almost at all frequency ranges. The best directivity is obtained at 4 GHz of 12.877 dBi. Figure 7b shows the directivity performance of single and 1 × 2 RPT, 1 × 2 RLPT, 1 × 2 ELPT, and 1 × 2 MLPT. At frequencies, less than 2.5 GHz the 1 × 2 ELPT got the best performance of directivity. However, at a frequency of more than 2.5 GHz, 1 × 2 MLPT has the best directivity. At 4 GHz, the directivity of 1 × 2 RPT, RLPT, ELPT, and MLPT is 12.108 dBi, 14.329 dBi, 15.248 dBi, and 16.561 dBi. It means there is an improvement of directivity of 4.453 dBi. Although the MLPT has a larger substrate size than regular ones, adding a lens and metamaterial structure can trap the electric field in the lens causing the gain increase.

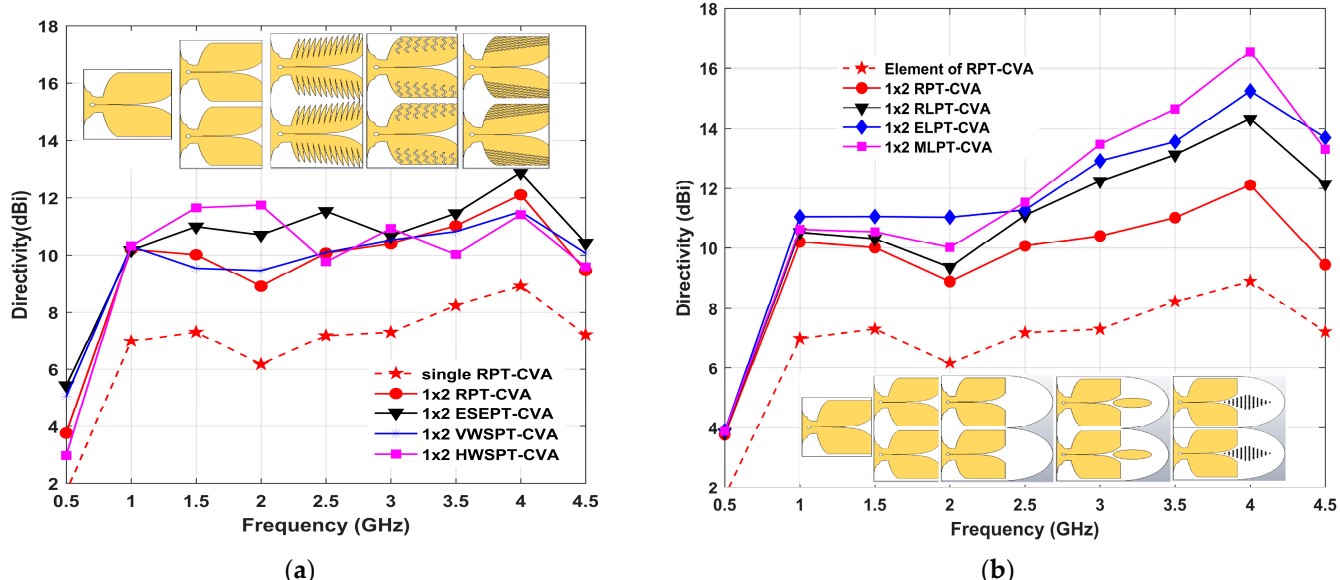

**(a)**                **(b)**

**Figure 7.** Directivity of (**a**). Element and 1 × 2 Regular Palm Tree- (RPT-CVA), 1 × 2 Exponential Slot Edge Palm Tree (ESE-CVA), 1 × 2 Vertical Wave Structure Palm Tree (VWPT-CVA), 1 × 2 Horizontale Wave Structure Palm Tree (HWPT-CVA), and (**b**). Element and 1 × 2 Regular Palm Tree-Coplanar Vivaldi Antenna (RPT-CVA), 1 × 2 Regular Lens Palm Tree (RLPT-CVA), 1 × 2 Elips Lens Palm Tree (ELPT-CVA), And 1 × 2 Metamaterial Lens Palm Tree (MLPT-CVA).

However, in a wideband antenna, the wider the antenna bandwidth, the greater distance between elements relative to their wavelength (especially at high-end frequencies), and this cause a grating lobe which will reduce the directivity of the antenna. The grating lobe is a side lobe that is enlarged and resembles the main lobe, this thing caused by the effect of changing the distance between the antenna elements further apart. In this case the 1 × 2 MIMO has spacing between element is 275 mm and it means that at a frequency of 0.5 GHz, the distance between elements is 0.458λ while at a frequency of 4.5 GHz the distance between elements is 4.125λ. The distance between elements rises at 4.5 GHz, causing the grating lobe and antenna directivity to diminish.

### 3.2.2. Side Lobe Level Performance

Figure 8a shows the Side Lobe Level (SLL) of element RPT, 1 × 2 of RPT, 1 × 2 of FCPT, MCPT, BCPT, CCPT, LTPT, RTPT, and HFSP. Meanwhile, Figure 8b presents the SLL of element RPT, SLL 1 × 2 of RPT, 1 × 2 of KSPT, ESEPT, VWSPT, HWSPT, RLPT, ELPT, and

MLPT-CVA. From the simulation result, It shows that the best SLL performance is reached for RPT of 11.94 dB at 1 GHz, followed by 1 × 2 HFSPT of 10.357 dB at 3 GHz. Furthermore, most antennas have an SLL greater than −5 dB at frequencies above 1.5 GHz. Although mutual scaring parameters of the antenna at high frequencies are good, it produces a high-level sidelobe. This is because the higher the frequency, the greater the distance between the elements relative to the wavelength, therefore, the grating lobe will occur, enlarging the SLL However the SLL of a single element of RPT is better than others at frequency 2, 2.5, 3.5 and 4 GHz. But at frequency 1.5 GHz shows that 1 × 2 RPT has better performance of SLL than a single element. For wideband antennas, by arranging the antennas into MIMO, the sidelobe level performance can increase at low frequencies but at high frequencies, the SLL performance decreases due to the presence of grating lobes because the distance between elements becomes greater. Figure 8b shows that the 1 × 2 ELPT got the best SLL performance at 1 GHz of −13.757 dB. At 2 GHz, the 1 × 2 KFSPT results in the best performance of SLL of −9.426 dB. Meanwhile, at 2 GHz, the worst SLL of −0.677 dB was obtained for RPT. This means an SLL improvement of 8.749 dB between 1 × 2 KFSPT and 1 × 2 RPT. The best SLL of −6.89 dB was obtained at 2.5 GHz for 1 × 2 MLPT, while at 4 GHz, the 1 × 2 ELPT has the best performance of −4.688 dB. SLL performance can also be improved by adding a structure and lens at both antenna heights.

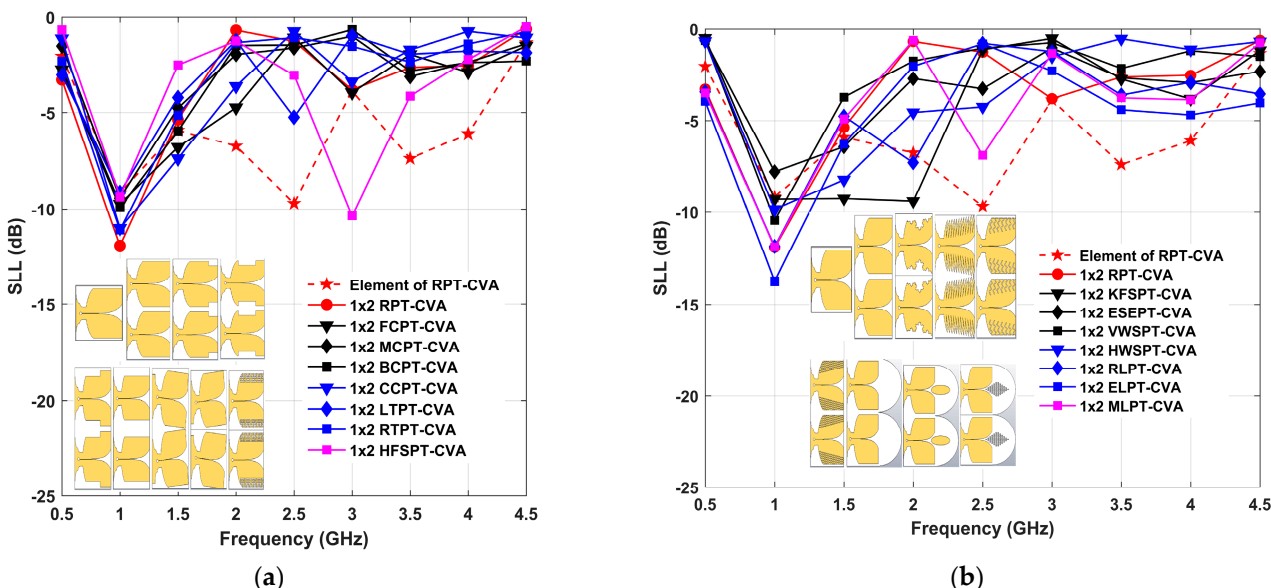

(**a**)  (**b**)

**Figure 8.** Side Lobe Level of (**a**). Element and 1 × 2 Regular Palm Tree (RPT-CVA), 1 × 2 Front Cut Palm Tree (FCPT-CVA), 1 × 2 Middle Cut Palm Tree (MCPT-CVA), 1 × 2 Back Cut Palm Tree (BCPT-CVA), 1 × 2 Complete Cut Palm Tree (CCPT-CVA), 1 × 2 Left Tilt Palm Tree (LTPT-CVA), 1 × 2 Right Tilt Palm Tree (RTPT-CVA), 1 × 2 Hilbert Fractal Structure Palm Tree (HFSPT-CVA), and (**b**). Element and 1 × 2 Regular Palm Tree (RPT-CVA), 1 × 2 Koch Fractal Structure Palm Tree (KFSPT-CVA), 1 × 2 Exponential Slot Edge Palm Tree (ESEPT-CVA), 1 × 2 Vertical Wave Structure Palm Tree (VWSPT-CVA), 1 × 2 Horizontale Wave Structure Palm Tree (HWSPT-CVA), 1 × 2 Regular Lens Palm Tree (RLPT-CVA), 1 × 2 Elips Lens Palm Tree (ELPT-CVA), and 1 × 2 Metamaterial Lens Palm Tree (MLPT-CVA).

### 3.2.3. Beam Squint and Beamwidth Performance

The boresight to 10 dB beamwidth ratio is known as beam squint. The boresight should be symmetrical if the beam squint is zero. The feeding network settings, the design of the patch or radiator, and the design of the antenna substrate can all determine the polarization of the antenna, which can influence the beam squint. Ideally, the beam squint is zero which means the boresight is symmetrical. Changes in the beam squint can be induced by changes in the polarization of the antenna, which can be caused by the feeding network settings,

the geometry of the patch/radiator, and the shape of the antenna substrate. The squint beam will reduce the link budget due to the misalignment of the main beam [42].

Figure 9a shows the beam squint of the element and $1 \times 2$ RPT, $1 \times 2$ of FCPT, MCPT, BCPT, CCPT, LTPT, RTPT, and HFSPT-CVA. The simulation results that HFSPT has the worst performance at 0.5 GHz and 2 GHz. At 2 GHz the result of beam squint is 32,425° and $-30.613°$ for $1 \times 2$ HFSPT and $1 \times 2$ RTPT respectively. Figure 9b describes the beam squint of the element and $1 \times 2$ of RPT, $1 \times 2$ of KFSPT, ESEPT, VWSPT, HWSPT, RLPT, ELPT, and MLPT-CVA. At 0.5 GHz $1 \times 2$ KFSPT has the worst beam squint as shown in Figure 9b. Adding a corrugated slot can affect the beam squint as well as the return loss performance in the low-end frequency. The $1 \times 2$ KFSPT also results in the beam squint of 32.87°, 25.44°, and 21.048° at 2 GHz, 2.5 GHz, and 3 GHz respectively. Figure 9b, found that ELPT results in the best beam squint performance, in all frequencies between 0.5 GHz to 4.5. However, the $1 \times 2$ RPT element maximum Beamsquint is 3.28° at frequencies 0.5 and 4 GHz. Adding structure on. both edges of the patch can increase the gain but also affect the beam squint of the antenna.

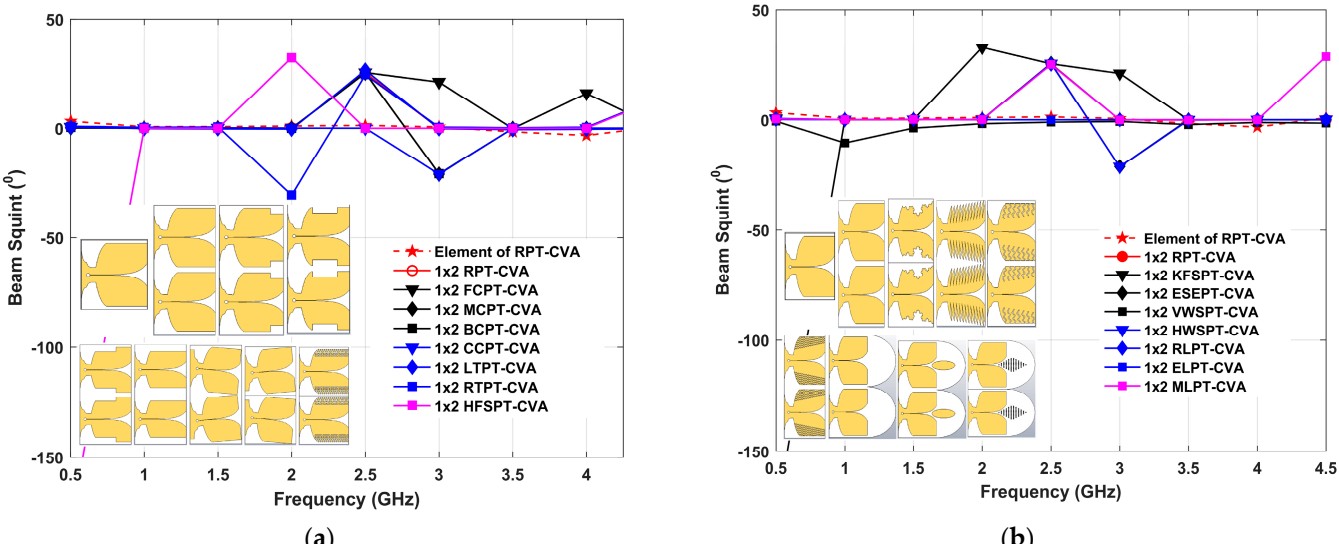

**(a)**      **(b)**

**Figure 9.** Beam Squint Performance of (**a**) Element and $1 \times 2$ Regular Palm Tree (RPT-CVA), $1 \times 2$ Front Cut Palm Tree (FCPT-CVA), $1 \times 2$ Middle Cut Palm Tree (MCPT-CVA), $1 \times 2$ Back Cut Palm Tree (BCPT-CVA), $1 \times 2$ Complete Cut Palm Tree (CCPT-CVA), $1 \times 2$ Left Tilt Palm Tree (LTPT-CVA), Right Tilt Palm Tree (RTPT-CVA), $1 \times 2$ Hilbert Fractal Structure Palm Tree (HFSPT-CVA), and (**b**). Element and $1 \times 2$ Regular Palm Tree (RPT-CVA), $1 \times 2$ Koch Fractal Structure Palm Tree (KFSPT-CVA), $1 \times 2$ Exponential Slot Edge Palm Tree (ESEPT-CVA), $1 \times 2$ Vertical Wave Structure Palm Tree (VWSPT-CVA), $1 \times 2$ Horizontale Wave Structure Palm Tree (HWSPT-CVA), $1 \times 2$ Regular Lens Palm Tree (RLPT-CVA), $1 \times 2$ Elips Lens Palm Tree (ELPT-CVA), and $1 \times 2$ Metamaterial Lens Palm Tree (MLPT-CVA).

Figure 10 shows that the antenna has a large beamwidth at a frequency of 0.5 GHz, which decreases with increasing frequency. The largest beamwidth obtained by the ELPT-CVA structure at 0.5 was 78.57°, while the smallest was by VWSPT-CVA at 41.41°. The antenna beamwidth at 4 GHz frequency shows the smallest beamwidth of 3.32° for the ELPT-CVA structure. Although at a frequency of 0.5 GHz, the ELPT-CVA antenna has the largest beamwidth, the addition of an elliptical structure, with increasing frequency, makes the beamwidth smaller. The beam width variation across the 15 simulated antenna in $1 \times 2$ MIMO configurations is modest at the 1–4.5 GHz frequency. Figure 10 illustrates that the single-element beamwidth is greater than the antenna in MIMO at all frequencies. Antennas in MIMO can increase beamwidth performance.

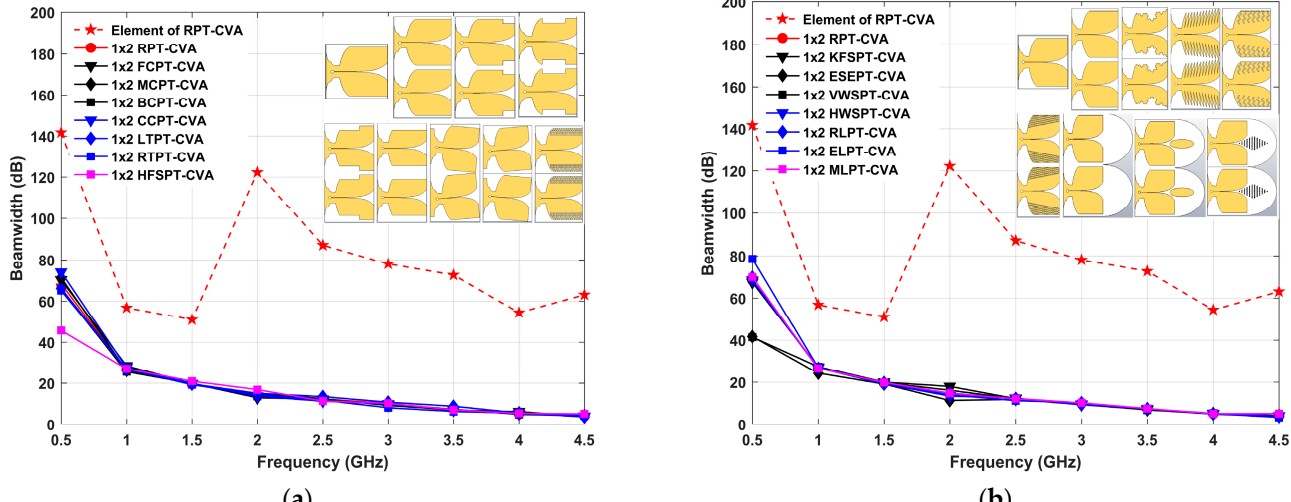

**Figure 10.** Beamwidth of (**a**). Element and 1 × 2 Regular Palm Tree (RPT-CVA), 1 × 2 Front Cut Palm Tree (FCPT-CVA), 1 × 2 Middle Cut Palm Tree (MCPT-CVA), 1 × 2 Back Cut Palm Tree (BCPT-CVA), 1 × 2 Complete Cut Palm Tree (CCPT-CVA), 1 × 2 Left Tilt Palm Tree (LTPT-CVA), 1 × 2 Right Tilt Palm Tree (RTPT-CVA), 1 × 2 Hilbert Fractal Structure Palm Tree (HFSPT-CVA), and (**b**). Element and 1 × 2 Regular Palm Tree (RPT-CVA), 1 × 2 Koch Fractal Structure Palm Tree (KFSPT-CVA), 1 × 2 Exponential Slot Edge Palm Tree (ESEPT-CVA), 1 × 2 Vertical Wave Structure Palm Tree (VWSPT-CVA), 1 × 2 Horizontale Wave Structure Palm Tree (HWSPT-CVA), 1 × 2 Regular Lens Palm Tree (RLPT-CVA), 1 × 2 Elips Lens Palm Tree (ELPT-CVA), and 1 × 2 Metamaterial Lens Palm Tree (MLPT-CVA).

### 3.2.4. Rectangular Radiation Characteristic

Figure 11 displays some of the outcomes of the RPT antenna's radiation pattern with a different slot structure. Figure 11a,b show the 2 GHz radiation patterns between the RPT, FCPT, and HWSPT antennas in the E-plane. Figure 11c,d show the radiation patterns comparison of the RPT to the ESEPT, and MLPT antennas at 4 GHz, respectively. At a frequency of 2 GHz, the main lobe RPT is 8.89 dBi, the side lobe level (SLL) is −0.7 dB, the Angular width (3 dB) is 15.2° and the main lobe direction is 0°. The FCPT structure results of 11.6 dBi main lobe, −4.7 dB sidelobe level, 14.1° angular widths, and 0° main lobe direction. While the HWSPT structure generates a major lobe of 11.7 dBi, the main lobe with an angular width of 14.6°, and the main lobe direction of 0°, as well as a sidelobe level of −4.6 dB. The antenna performance of the FCPT and HWSPT structures at 2 GHz is superior to RPT in the main lobe, SLL, and Angular width (3 dB). For instance, At the frequency of 4 GHz, ESEPT generates the main lobe of 12.9 dBi, the main lobe direction of 0°, an angular width of 7.4°, and a side lobe level of −2.5 dB. At the frequency of 4 GHz, the performance of the RPT and ESEPT antenna radiation is almost the same, with only an increase in the directivity of 0.8 dBi. At 4 GHz, the MLPT produces a main lobe of 16.6 dBi, a Mainlobe direction of 0°, an angular width of 23.1°, and a side lobe level of −11.1 dB. The directivity of MLPT increased by 4.5 dBi due to the addition of the lens structure and metamaterial so that electromagnetic waves are embedded in the metamaterial structure.

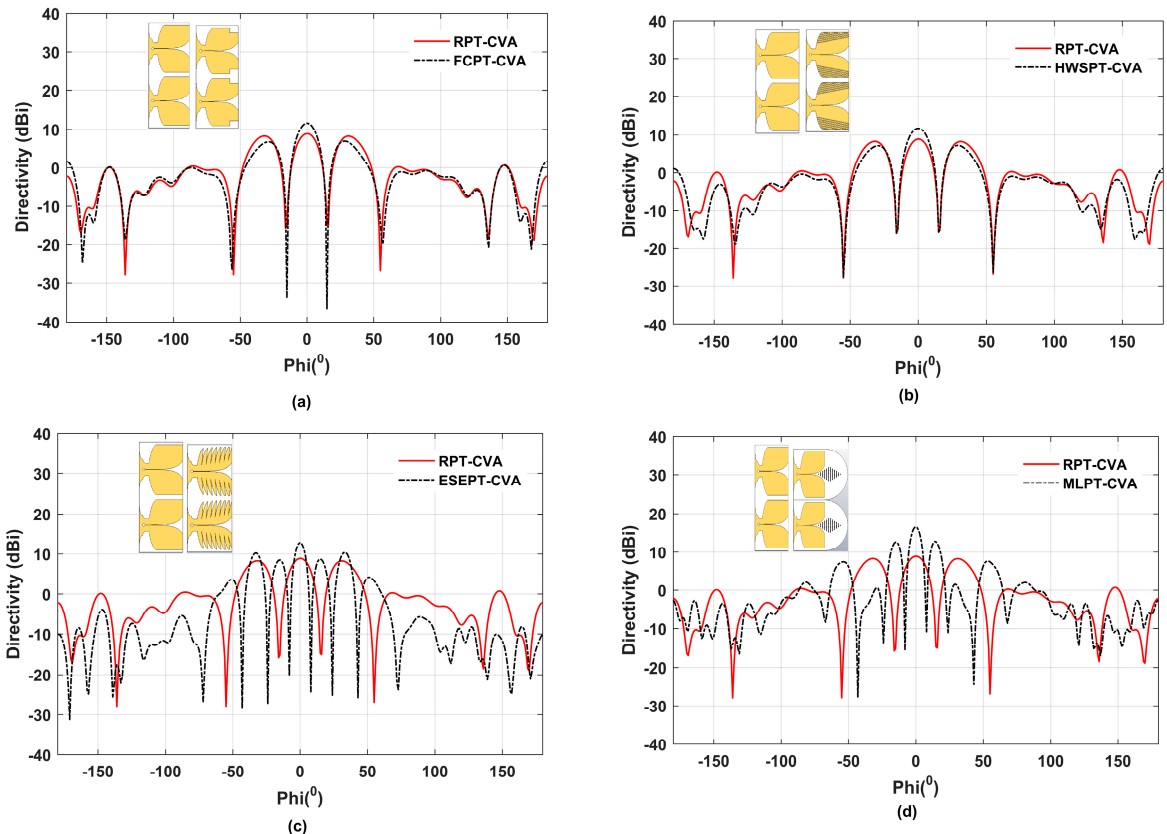

**Figure 11.** Radiation Pattern in the E-Plane of (**a**). RPT-CVA vs FCPT-CVA at 2 GHz, (**b**). RPT-CVA vs HFSPT-CVA at 2 GHz, (**c**) RPT-CVA vs ESE-CVA at 4 GHz, and (**d**). RPT-CVA vs MLPT-CVA at 4 GHz.

### 3.3. Surface Current Performance

Surface current is an electric current induced by an electromagnetic field. The surface current distribution varies with frequency. Figure 12 depicts the surface current of several antennas at a frequency of 0.5 GHz while Figure 13 shows the surface current of 15 antennas at 2 GHz.

In this case, we set the maximum surface current of all antennas at 0.5 A/m. Surface current varies for all antenna types, as seen in Figures 12 and 13. We designed the antenna by providing a distance between adjacent patches (copper) of 25 mm (as shown in Figure 1b) so that the surface current does not flow directly to adjacent patch elements (copper radiators) when the antennas are placed closely together. At 0.5 GHz, even though the copper radiators are separated by 25 mm, there is still a dispersion of surface currents with high intensity in nearby elements (indicated in red in the yellow circle with dotted lines) as demonstrated in Figure 12a,b. RPT and FCPT had larger surface currents in both neighboring patches than BCPT. This demonstrates that BCPT outperforms the rest in terms of S11 performance. There are various places with high surface current intensity in slots such as HFSPT, ESEPT, VWSPT, and HWSPT that are highlighted in red. Figure 12 shows that at a frequency of 0.5 GHz, the surface current in the elliptical structure and metamaterial is not excessive. Antennas with a lens structure that has been given an elliptical slot structure or metamaterial show increased surface current concentration at 4 GHz. A significant electric field is described by the existence of a high surface current.

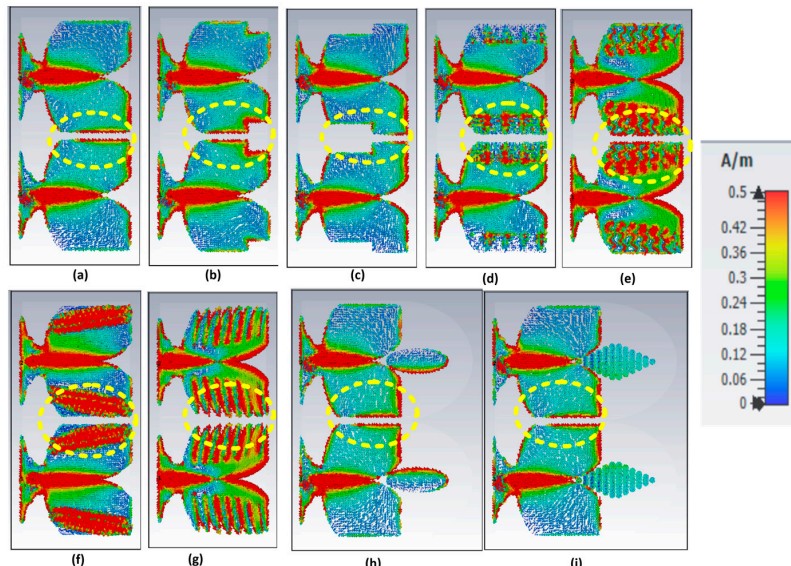

**Figure 12.** Surface current performance of (**a**) Regular Palm Tree (RPT-CVA), (**b**) Front Cut Palm Tree (FCPT-CVA), (**c**) Back Cut Palm Tree (BCPT-CVA), (**d**) Hilbert Fractal Structure Palm Tree (HFSPT-CVA), (**e**) Vertical Wave Structure Palm Tree (VWSPT-CVA), (**f**) Horizontale Wave Structure Palm Tree (HWSPT-CVA), (**g**) Exponential Slot Edge Palm Tree (ESEPT-CVA), (**h**) Elips Lens Palm Tree (ELPT-CVA), and (**i**) Metamaterial Lens Palm Tree (MLPT-CVA).

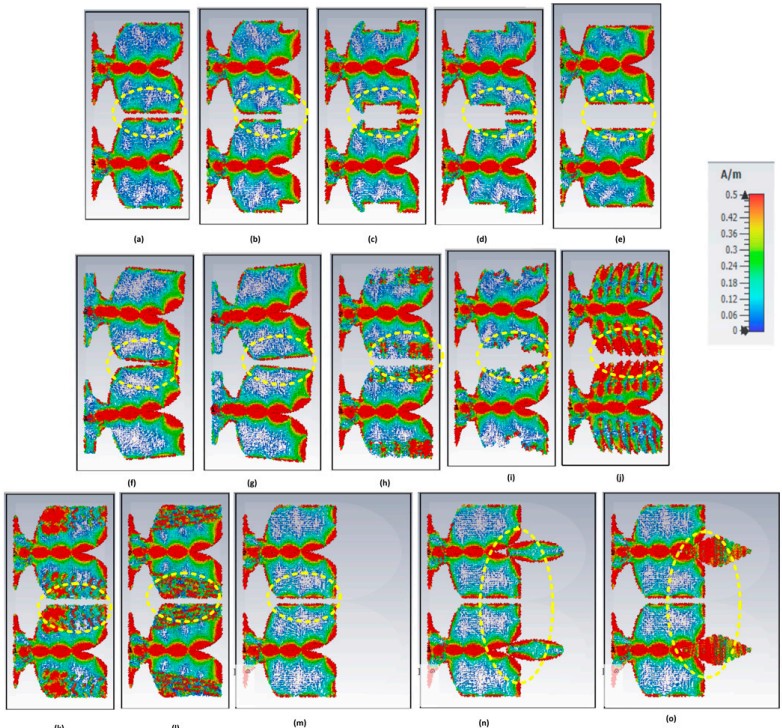

**Figure 13.** Surface current performance of (**a**) Regular Palm Tree (RPT-CVA), (**b**) Front Cut Palm Tree (FCPT-CVA), (**c**) Middle Cut Palm Tree (MCPT-CVA), (**d**) Back Cut Palm Tree (BCPT-CVA), (**e**) Complete Cut Palm Tree (CCPT-CVA), (**f**) Left Tilt Palm Tree (LTPT-CVA), (**g**) Right Tilt Palm Tree (RTPT-CVA), (**h**) Hilbert Fractal Structure Palm Tree (HFSPT-CVA), (**i**) Koch Fractal Structure Palm Tree (KFSPT-CVA), (**j**) Exponential Slot Edge Palm Tree (ESEPT-CVA), (**k**) Vertical Wave Structure Palm Tree (VWSPT-CVA), (**l**) Horizontale Wave Structure Palm Tree (HWSPT-CVA), (**m**) Regular Lens Palm Tree (RLPT-CVA), (**n**) Elips Lens Palm Tree (ELPT-CVA), and (**o**) Metamaterial Lens Palm Tree (MLPT-CVA).

## 4. Measurement and Comparison of Related Antenna

Figure 14 depicts the fabrication and comparison of measurement and simulation results for ESEPT-CVA and MLPT-CVA. At 4 GHz, both antennas have a greater gain than the RPT-CVA, as seen in Figure 7. Antenna measurements are carried out by taking $S_{11}$ antenna data using a brand VNA Siglent which works from a frequency of 100 kHz to 3.2 GHz. From the measurement results, it is found that the $S_{11}$ antenna measured by VNA produces better results at low frequencies. It is known that the measurement results are in agreement with the simulation results where most of the antennas have $S_{11}$ below $-10$ dB. High-gain antennas can be applied to radar applications.

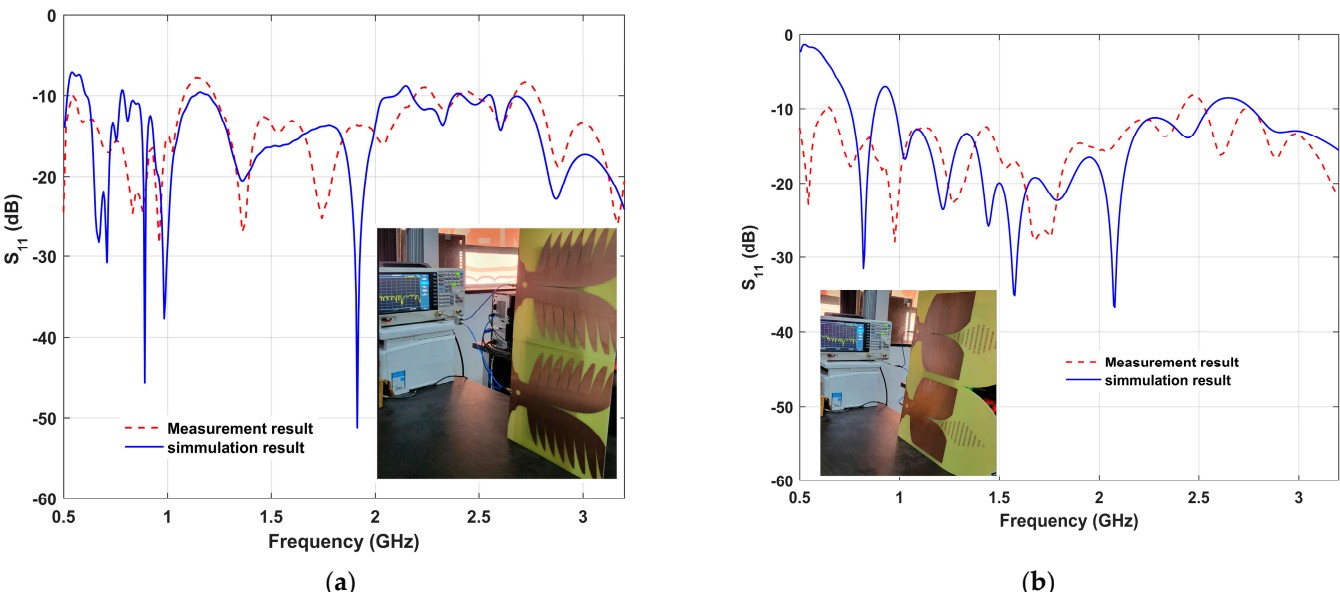

**(a)**                                                                                                          **(b)**

**Figure 14.** Simulation and measurement result of (**a**) ESEPT-CVA and (**b**) MLPT-CVA.

The Vector Network Analyzer is a tool that can use for measuring radio frequency scattering properties in radar applications. When two ports of the VNA are linked to the antenna, the $S_{21}$ data can explain the transfer function of the signals emitted and received as shown in Equations (4) and (5). $S_x(t)$ represents the chirp signal with a certain period and bandwidth, whereas $S_y(t)$ represents the received signal chirp [49].

$$S_y(f) = S_{21}S_x(f) \tag{4}$$

$$S_r(t) = F^{-1}S_y(f) \tag{5}$$

In this study, we used an antenna to detect objects behind the wall. The antenna used is MLPT by connecting it to a portable nano VNA that works at a frequency of 0.5–3 GHz. The antenna and the detected object are varied in distance to the wall. The detected object is a laptop varying the object's distance from the wall and the detection process is carried out by placing the antenna on the E-plane. The antenna is connected to the VNA and the VNA is connected to the laptop. The Scattering signal received will be seen on the laptop display as shown in Figure 15.

The detected wall has an area of $60 \times 150$ cm$^2$ by dividing the area into several segments area in the $x$ and $y$ direction. The $S_{21}$ data received on the laptop will be processed and a signal reconstruction process will be carried out so that object detection results are produced as shown in Figure 16. The yellow circle shows the position of the detected target in the x-y plane.

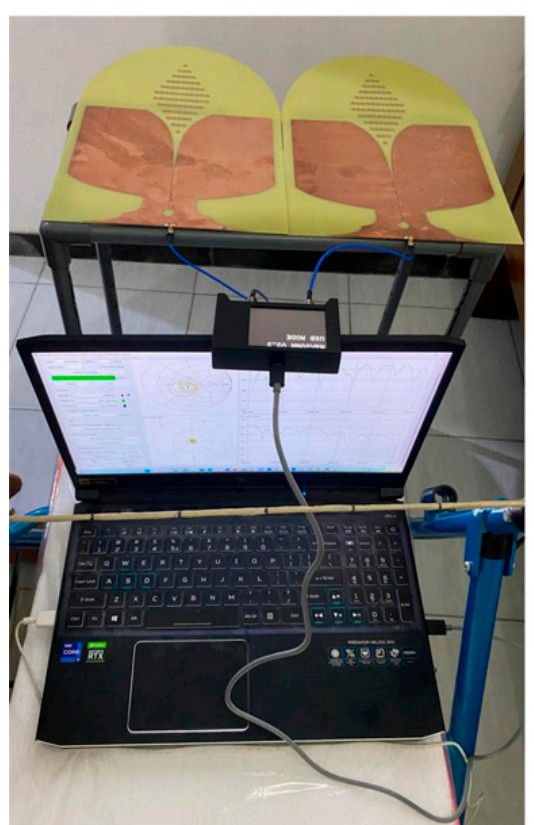
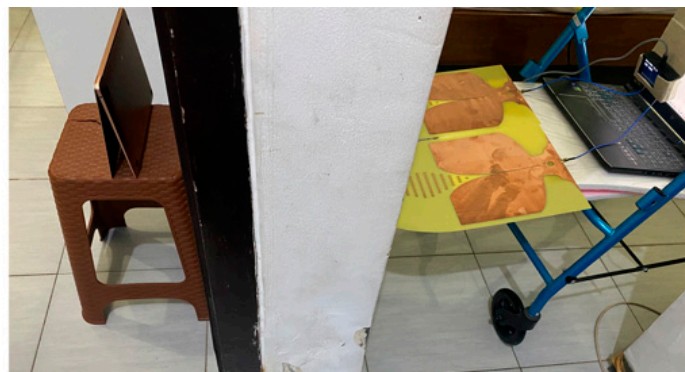
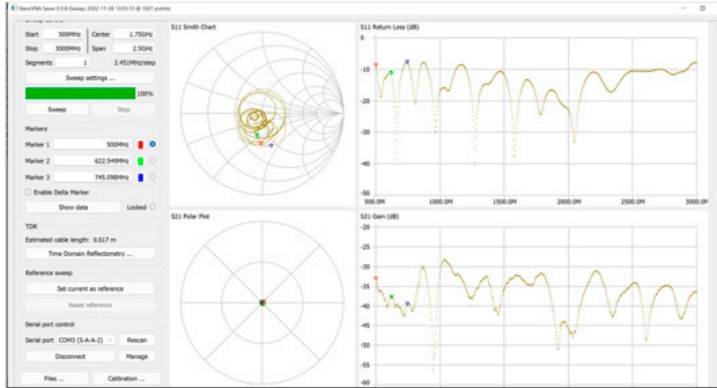

**Figure 15.** Radar target measurement with MLPT-CVA prototype.

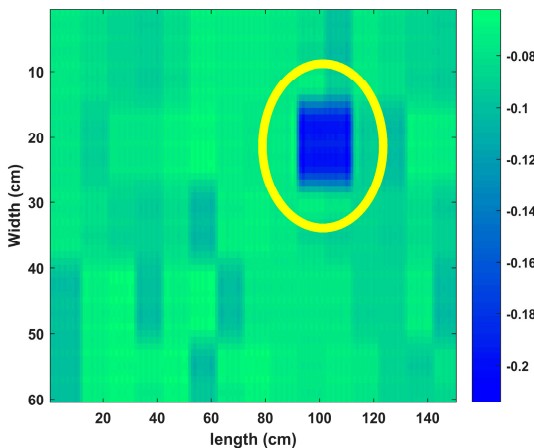

**Figure 16.** Radar target detection in the xy-planes.

Table 2 reveals that there has been some research on Vivaldi antennas that operate at low frequencies, but they have not taken into account mutual coupling. In this case, we also compare the antenna element. As shown in Table 2, Our antenna has a smaller width than the antenna at [50]. It employs ceramic materials to operate at a frequency of 0.5 to 3 GHz. The antenna has a wide bandwidth of 0.5–6 GHz has been discussed in [51]. That antenna achieves a gain of 8 dBi at a frequency of 2 GHz, but as frequency increased, the gain declined. A big CVA antenna, which operated at a very low frequency has been evaluated in [52]. However, the maximum frequency used in research [53,54] is 2.1 GHz. The AVA antenna for GPR application with metamaterial and DGS structure has been discussed in [55,56] with a larger size than our purposes study. Aspects of our research operate between 0.5 and 4.5 GHz. The ESEPT-CVA and MLPT-CVA antennas, with

respective elemental gains of 9.7 dBi and 13.4 dBi at 4 GHz, are used in this comparison. In this study, ESEPT-CVA and MLPT-CVA were arranged in MIMO $1 \times 2$ to produce 12.87 dBi and 16.561 dBi gains at 4 GHz frequency.

**Table 2.** Comparison of the proposed antenna and related research.

| Ref | Element Dimension (mm) | Ant. Type | Sub. Type | Freq. (GHz) | Gain (dBi) |
|---|---|---|---|---|---|
| [50] | 300 × 360 | AVA | T-ceramic | 0.5–3 | - |
| [51] | 258 × 150 | CVA | FR-4 | 0.5–6 | 8 |
| [52] | 950 × 780 | CVA | - | 0.02–0.12 | - |
| [53] | 260 × 185 | AVA | Taconic | 0.5–2 | - |
| [54] | 286 × 300 (with metamaterial) | CVA | FR4 | 0.7–2.1 | 10.5 |
| [55] | 450 × 600 | AVA | Rogers 4350 | 0.3–2 | 4.4–11.5 |
| [56] | 750 × 525 | CVA | FR4 | 0.26–0.34 | 4.2 |
| ESEPT | 275 × 275 | CVA | FR4 | 0.5–4.5 | 9.7 |
| MLPT | 275 × 438 | CVA | FR4 | 0.5–4.5 | 13.4 |

The comparative performance of 15 Vivaldi antennas is displayed in Table 3 to get the performance in $1 \times 2$ Palm Tree MIMO antenna. Table 3 shows that the best $S_{11}$ at the 0.5 GHz frequency was obtained by ESEPT which is $-16.86$ dB, but for the overall working frequency included in $S_{11}$ less than $-10$ dB is BCPT as shown in Figure 3b. Table 3 also shows that the best $S21$ at the 0.5 GHz frequency was obtained by HWSPT which was $-18.57$ dB. Table 3 shows that at 0.5 GHz, practically all models still have S21 > $-20$ dB because the distance between components is less than 0.5, implying that another strategy for mutual coupling reduction is required. Table 3 further shows that the maximum directivity attained by MLPT is 16.56 dBi, whereas ELPT produced the lowest SLL, best beam squint, and lowest beam width.

**Table 3.** $1 \times 2$ MIMO palm tree Coplanar Vivaldi Antena.

| Ant. Type | S11 (dB) At 0.5 GHz | S21 (dB) At 0.5 GHz | Max Dir (dB) | Min SLL (dB) | Max Beamsquit (°) (0.5–4.5 GHz) | Min Beamwidth (°) |
|---|---|---|---|---|---|---|
| RPT | −8.33 | −13.45 | 12.11 (4 GHz) | −11.94 (1 GHz) | 25.46 (2.5 GHz) | 3.98 (4.5 GHz) |
| FCPT | −7.29 | −13.73 | 11.58 (2 GHz) | −9.73 (1 GHz) | 25.84 (2.5 GHz) | 3.78 (4.5 GHz) |
| MCPT | −10.04 | −15.19 | 12.09 (4 GHz) | −9.43 (1 GHz) | 25.50 (2.5 GHz) | 4.41 (4.45 GHz) |
| BCPT | −10.48 | −14.06 | 12.10 (4 GHz) | −9.9 (11 GHz) | 25.46 (2.5 GHz) | 4.27 (4.5 GHz) |
| CCPT | −6.57 | −16.4 | 11.19 (4 GHz) | −11.02 (1 GHz) | −21.17 (3 GHz) | 3.99 (4.5 GHz) |
| LTPT | −9.36 | −11.49 | 11.75 (4 GHz) | −9.16 (1 GHz) | 26.69 (2.5 GHz) | 3.64 (4.5 GHz) |
| RTPT | −10.37 | −15.38 | 11.50 (3.5 GHz) | −11.11 (1 GHz) | −30.63 (2 GHz) | 4.4 (4.5 GHz) |
| HFSP | −12.21 | −14.5 | 12.06 (3.5 GHz) | −10.35 (1 GHz) | −179.62 (0.5 GHz) | 5.22 (4.5 GHz) |
| KFSPT | −14.33 | −15.29 | 12.68 (4 GHz) | −9.43 (2 GHz) | −9.43 (2 GHz) | 4.09 (4.5 GHz) |
| ESEPT | −16.86 | −13.43 | 12.87 (4 GHz) | −7.79 (1 GHz) | −179.81 (0.5 GHz) | 4.49 (4.5 GHz) |
| VWSPT | −15.63 | −13.73 | 11.51 (4 GHz) | −10.46 (1 GHz) | −179.82 (0.5 GHz) | 3.69 (4.5 GHz) |
| HWSPT | −6.53 | −18.57 | 11.74 (2 GHz) | −9.57 (1 GHz) | 25.53 (2.5 GHz) | 4.09 (4.5 GHz) |
| RLPT | −8.71 | −12.57 | 14.43 (4 GHz) | −11.89 (1 GHz) | 25.63 (2 GHz) | 3.79 (4.5 GHz) |
| ELPT | −8.92 | −12.53 | 15.25 (4 GHz) | −13.77 (1 GHz) | 0 | 3.32 (4.5 GHz) |
| MLPT | −8.87 | −12.74 | 16.56 (4 GHz) | −11.91 (1 GHz) | 29.27 (4.5 GHz) | 5.22 (4.5 GHz) |

## 5. Conclusions

We have simulated 15 kinds of palm tree antennas in the 0.5–4.5 GHz frequency with several structures, namely Regular Palm Tree-Coplanar Vivaldi Antenna (RPT-CVA), Front Cut Palm Tree (FCPT-CVA), Middle Cut Palm Tree (MCPT-CVA), Back Cut Palm Tree (BCPT-CVA), Complete Cut Palm Tree (CCPT-CVA), Left Tilt Palm Tree (LTPT-CVA), Right Tilt Palm Tree (RTPT-CVA), Hilbert Fractal Structure Palm Tree (HFSPT- CVA), Koch Fractal Structure Palm Tree (KFSPT-CVA), Exponential Slot Edge Palm Tree (ESEPT-CVA), Vertical

Wave Structure Palm Tree (VWPT-CVA), Horizontal Wave Structure Palm Tree (HWPT-CVA), Regular Lens Palm Tree (RLPT-CVA), Elliptical Lens Palm Tree (ELPT-CVA) and Metamaterial Lens Palm Tree (MLPT-CVA). By giving a different structure to the antenna radiator while maintaining the same substrate width and feed parameters and the similarity of the slope of the two tapered slots, the performance of return loss, mutual scaring, beam squint, and beamwidth produce different performances. From the simulation results, it is found that HWSPT gets the best mutual scaring performance at low frequencies because it has less than $-10$ dB mutual scaring. The maximum directivity of the RPT-CVA is 12.108 dBi, while the MLPT-CVA has a gain of 16.561 dBi at the 4 GHz frequency. The beam squint at all frequencies is $0°$ for ELPT-CVA and the lowest beamwidth is also obtained by ELPT-CVA at 4.5 GHz. This comparative analysis can be used as a reference for the selection of MIMO antenna design in considering the performance requirements of return loss, mutual coupling, directivity, beam squint, and beam width. From the results of the return loss measurements, there is also a match between the simulation results and antenna measurements where the antenna can work at a frequency of 0.5–4.5 GHz so that this antenna can be recommended for radar applications.

**Author Contributions:** Conceptualization, N.N. and A.M.d.O.; methodology, E.S.; software, N.N. and D.K.; validation, N.N. and E.S.; formal analysis, N.N. and M.N.M.Y.; investigation, D.K.; resources, E.S.; data curation, N.N. and D.K.; writing—original draft preparation, N.N.; writing—review and editing, E.S. and M.N.M.Y.; supervision, A.M.d.O.; project administration, N.N.; funding acquisition, N.N. and M.N.M.Y. All authors have read and agreed to the published version of the manuscript.

**Funding:** We appreciate the funds granted by the Directorate of Research, Technology, and Community Service under Decree No. 313/UN38/HK/PP/2022 and Agreement/Contract No. 039/E5/PG.02.00.PT/2022 and the international grant UNESA-UNIMAP (INTERES).

**Institutional Review Board Statement:** Not applicable.

**Data Availability Statement:** Not applicable.

**Conflicts of Interest:** The authors declare no conflict of interest.

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
