# Peer review of "Design of 1 × 2 MIMO Palm Tree Coplanar Vivaldi Antenna in the E-Plane with Different Patch Structure"

_electronics, doi:10.3390/electronics12010177_

Round 1

Reviewer 1 Report

The authors presented an interesting work. Please consider the following comments that need to be addressed carefully to make the final decision:

Please revise the title of the manuscript as from the word "performance comparison" it sounds like a review article. 

The title needs to be revised carefully to clearly show the purpose of the study, from the title it seems like a review article.

The abstract does not convey the main contribution of the article, designing so many antennas is useless without a proper problem statement. Thus abstract should be rewritten carefully to explain the problem statement.

In lines 11-12, what does it refer to in the evaluation of an antenna in an E-plane????

Throughout the manuscript, a lot of typos are found that need to be revised carefully.

a) In line 23: what does 3,32 refers to? 

b) keywords are not in alphabetic order

c) avoid using the same term multiple times in keywords, like Vivaldi.

More effort should be put to cite the state of the art. It is not a good way to cite a huge number of conference papers, thus, papers from the state of the art journals like MDPI, IEEE or Elsevier should be included. Having a better literature study increases the interest of the readers. On-Demand frequency switchable antenna array operating at 24.8 and 28 GHz for 5G High-Gain sensors applications or isolation improvement of parasitic element-loaded dual-band MIMO antenna for mm-wave applications or design and analysis of a simple miniaturized fractal antenna for 5G ka-band applications can be utilized to replace the ref 1- 5.

 Equations must be cited using appropriate references.

The quality of the figures needs to improve, the symbols and legends are not readable in most cases. Fig. 1,2,11, and 14 need to revise more carefully.

It will be better to add various types of antennas in literature and present the advantages of Vivaldi antenna to better catch the attention of the readers.

Author Response

Response to Reviewer 1 Comments

Comments and Suggestions for Authors

The authors presented an interesting work. Please consider the following comments that need to be addressed carefully to make the final decision:

Point 1: Comments to the Author:

Please revise the title of the manuscript as from the word "performance comparison" it sounds like a review article. The title needs to be revised carefully to clearly show the purpose of the study, from the title it seems like a review article.

Author's response:

Thank you for directing us to revise the title of the manuscript. We have revised the title :

Design of 1×2 Palm Tree Coplanar Vivaldi Antenna Array in the E-Plane with Different Patch Structure

Point 2: Comments to the Author:

The abstract does not convey the main contribution of the article, designing so many antennas is useless without a proper problem statement. Thus abstract should be rewritten carefully to explain the problem statement.

Author's response:

Thank you for your comment and feedback. We have added a problem statement in the abstract as shown :

In this paper, 1×2 of Palm Tree Coplanar Vivaldi Antenna Arrays is presented that simulated at 0.5-4.5 GHz. Some GPR applications require wideband antennas starting from a frequency below 1 GHz to overcome high material loss and achieve deeper penetration. However, to boost the gain, antennas are set up in an array and this is costly due to the large size of the antenna. When configuring an array antenna in the E-plane, there is occasionally uncertainty over which antenna model may provide the optimum performance in terms of return loss, mutual coupling, directivity, beam squint, beam width, and surface current using a given substrate size. However, the configuration of E-plane antenna arrays has an issue of mutual coupling if the distance between elements is less than 0.5λ. Furthermore, it produces grating lobes at high frequencies. We implement several types of patch structures by incorporating the truncated, tilt shape, Hlbert and Koch Fractal, Exponential slot, Wave slot, the lens with elips, and metamaterial slot to the radiator by keeping the width of the substrate and the shape of the feeder. The return loss, mutual coupling, directivity, beam squint, beamwidth, and surface current of the antenna are compared for 1×2 MIMO CVA. A continuous patch array has a spacing of 0.458λ0 at 0.5 GHz, which is equivalent to its element width. From the simulation, we found that Back Cut Palm Tree (BCPT) and Horizontale Wave Structure Palm Tree (HWSPT) got the best performance of return loss and mutual scattering at low-end frequency respectively. The improvement of directivity got for Metamaterial Lens Palm Tree (MLPT) of 4,453 dBi if compared with Regular Palm Tree-Coplanar Vivaldi Antena Array (RPT) at 4GHz. Elips Lens Palm Tree (ELPT) has the best beam squint performance across all frequencies of 00. It also gots the best beamwidth at 4.5 GHz of 3,320. In addition, we incorporate the MLPT into the radar application.

Point 3: Comments to the Author:

In lines 11-12, what does it refer to in the evaluation of an antenna in an E-plane????

Author's response:

We have changed and rewritten the abstract as the reviewer suggested (based on the previous reviewer's comment). We have deleted the word evaluation in lines 11-12.

In the previous, the evaluation of an antenna in an E-plane means that we design and evaluate the performance of 15 models of 1x2 Palm tree Coplanar Vivaldi Antenna. The antenna's performance, includes return loss, mutual coupling, directivity, beam squint, beamwidth, and surface current. E- Plane is the electric field plane. So the palm tree antenna 1×2 is arranged parallel to the direction of the electric field

Point 4: Comments to the Author:

Throughout the manuscript, a lot of typos are found that need to be revised carefully.

  1. a) In line 23: what does 3,32 refer to? 
  2. b) keywords are not in alphabetic order
  3. c) avoid using the same term multiple times in keywords, like Vivaldi.

More effort should be put to cite the state of the art. It is not a good way to cite a huge number of conference papers, thus, papers from the state of the art journals like MDPI, IEEE, or Elsevier should be included. Having a better literature study increases the interest of the readers. On-Demand frequency switchable antenna array operating at 24.8 and 28 GHz for 5G High-Gain sensors applications or isolation improvement of parasitic element-loaded dual-band MIMO antenna for mm-wave applications or design and analysis of a simple miniaturized fractal antenna for 5G ka-band applications can be utilized to replace the ref 1- 5.

Author's response:

Thank you for all your input on the manuscript improvement.

  1. a) In line 23: The 3,320 refers to the beamwidth of ELPT at 4.5 GHz. ELPT produces the smallest beamwidth compared to other models. It states n the abstract line 30: “It also gots the best beamwidth at 4.5 GHz of 3,320

b). We have revised and ordered the keyword based on alphabetic:

Bandwidth, Coplanar Vivaldi Antenna, Mutual Coupling, Radiation Pattern.

  1. c) We have revised the keyword and removed the multiple times in keywords, like Vivaldi.

Bandwidth, Coplanar, Mutual Coupling, Radiation Pattern, Vivaldi Antenna

We have changed the conference paper in the reference from a reputable journal in [1]-[5] as the reviewer suggests. We add “On-Demand frequency switchable antenna array operating at 24.8 and 28 GHz for 5G High-Gain sensors applications in [1] and isolation improvement of parasitic element-loaded dual-band MIMO antenna for mm-wave applications in [32] as advised by the reviewer.

[1]         W. A. Awan, M. Soruri, M. Alibakhshikenari, and E. Limiti, “On-Demand Frequency Switchable Antenna Array Operating at 24.8 and 28 GHz for 5G High-Gain Sensors Applications,” Prog. Electromagn. Res. M, vol. 108, no. January, pp. 163–173, 2022, doi: 10.2528/PIERM21121102.

[2]         C. T. Mohamadi, M. Asefi, S. Thakur, J. Paliwal, and C. Gilmore, “Improved Metallic Enclosure Electromagnetic Imaging Using Ferrite Loaded Antennas,” Electron., vol. 11, no. 22, pp. 1–21, 2022, doi: 10.3390/electronics11223804.

[3]         S. M. Aguilar, M. A. Al-Joumayly, M. J. Burfeindt, N. Behdad, and S. C. Hagness, “Multiband miniaturized patch antennas for a compact, shielded microwave breast imaging array,” IEEE Trans. Antennas Propag., vol. 62, no. 3, pp. 1221–1231, 2014, doi: 10.1109/TAP.2013.2295615.

[4]         R. Guo, Y. Ni, H. Liu, F. Wang, and L. He, “Signal Diverse Array Radar for Electronic Warfare,” IEEE Antennas Wirel. Propag. Lett., vol. 16, pp. 2906–2910, 2017, doi: 10.1109/LAWP.2017.2751648.

[5]         S. Lee, S. Kim, Y. Park, and J. Choi, “A 3D-Printed tapered cavity-backed flush-mountable ultra-wideband antenna for UAV,” IEEE Access, vol. 7, pp. 156612–156619, 2019, doi: 10.1109/ACCESS.2019.2949795

[32]     M. Hussain et al., “Isolation Improvement of Parasitic Element-Loaded Dual-Band MIMO Antenna for Mm-Wave Applications,” Micromachines, vol. 13, no. 11, p. 1918, 2022, doi: 10.3390/mi13111918.

Point 5: Comments to the Author:

 Equations must be cited using appropriate references.

Author's response:

We have provided relevant references to all of the equations in the manuscript as shown in lines 152

The beginning and ending points of an exponential curve are x1, y1, x2, and y2[47].

 Figure 1(h) delivers the Hilbert curve structure in the third iteration with the total length of the slot (lh), the line segment (dd), and iteration(in) follows equation (2)[47]. In this case we use 3rd iteration.

Figures 2(c) and (d) show the vertical and horizontal wave slots. The Constanta of wavy slot based on equation (3) In this case, we set the B1= 2, B2= 1, B3= 1, B4= 5, and B5= 36. The depth of the wave, the number of waves, and the length of the wave slot can all be modified by changing the value of Bn[48]

 When two ports of the VNA are linked to the antenna, the S21 data can explain the transfer function of the signals emitted and received as shown in equations (4) and (5). Sx(t) represents the chirp signal with a certain period and bandwidth, whereas Sy(t) represents the received signal chirp[49].

Point 6: Comments to the Author:

The quality of the figures needs to improve, the symbols and legends are not readable in most cases. Fig. 1,2,11, and 14 need to revise more carefully.

Author's response:

Thank you for the input. In the revised manuscript, we have revised Figure 1 (line 164), Figure 2 (Inline 170), Figure 11(In line 414), and Figure 14 as shown in line 464.

We apologize for Figure 14 is clearly visible      

Point 7: Comments to the Author:

It will be better to add various types of antennas in literature and present the advantages of the Vivaldi antenna to better catch the attention of the readers.

Author's response:

Thank you for all of your input to the manuscript's enhancement. We have included several types of the antenna as shown in the revised article as stated in lines 86-90:

“Many different types of antennas may be used in radar and communications applications, including patch antennas[3], monopole antennas[6][8], and 3D antennas[5]. However, some of those antennas have an omnidirectional radiation pattern, or if they do have a directional radiation pattern, the directivity is minimal. Vivaldi antenna has advantages such as planar antenna, wide bandwidth, and directional radiation pattern”.

[3]         S. M. Aguilar, M. A. Al-Joumayly, M. J. Burfeindt, N. Behdad, and S. C. Hagness, “Multiband miniaturized patch antennas for a compact, shielded microwave breast imaging array,” IEEE Trans. Antennas Propag., vol. 62, no. 3, pp. 1221–1231, 2014, doi: 10.1109/TAP.2013.2295615.

[5]         S. Lee, S. Kim, Y. Park, and J. Choi, “A 3D-Printed tapered cavity-backed flush-mountable ultra-wideband antenna for UAV,” IEEE Access, vol. 7, pp. 156612–156619, 2019, doi: 10.1109/ACCESS.2019.2949795.

[6]         H. H. M. Ghouz, M. F. Aboee, and M. Aly Ibrahim, “Novel wideband microstrip monopole antenna designs for WiFi/LTE/WiMax devices,” IEEE Access, vol. 8, pp. 9532–9539, 2020, doi: 10.1109/ACCESS.2019.2963644.

[8]         N. Nurhayati, A. M. De-Oliveira, W. Chaihongsa, B. E. Sukoco, and A. K. Saleh, “A comparative study of some novel wideband tulip flower monopole antennas with modified patch and ground plane,” Prog. Electromagn. Res. C, vol. 112, pp. 239–250, 2021, doi: 10.2528/PIERC21040707.

Reviewer 2 Report

The authors discuss in detail the analysis and evaluation of the performance of fifteen Palm Tree Coplanar Vivaldi Antenna (PT-CVA) antennas in the E-plane, which was defined at a frequency in the range of f=0.5-4.5 GHz.  The classical designs were complemented by the implementation of several types of patch structures by incorporating truncated, tilted shape, Hlbert and Koch fractal, exponential slit, wave slit, lens with ellipses and metamaterial slit into the radiator while maintaining basic parameters such as substrate width and feed shape.
Back attenuation, mutual coupling, directivity, beam skew, beam width and surface current were analyzed and evaluated.
Based on simulations, analysis and evaluation, the authors found that the Back Cut Palm Tree (BCPT) and Horizontal Palm Tree (HWSPT) antennas achieved the best return loss and optimum performance, mutual dispersion at low frequency. A directivity improvement of 4.453 dBi occurs for the Metamaterial Lens Palm Tree (MLPT) compared to the Regular Palm Tree-Coplanar Vivaldi Antenna Array (RPT) at f=4 GHz. Furthermore, the Elips Lens Palm Tree (ELPT) antenna type has the best beam-narrowing performance in all cases and also has the best beamwidth at f=4.5 GHz.
The work is timely and interesting, makes a contribution to the field and contains elements of novelty. It is thorough and may be of value to the reader.
From a formal point of view, I recommend to revise and complete the description of the constants, variables in relations (1)-(3). I recommend to combine the numerical description of the geometry with the description of the geometry parameters in figures, Chapter 2.
No plagiarism was found.

Author Response

Electronics (MDPI)

Editor in Chief

Dear Editor and Reviewer

We would like submit our revised manuscript entitled, “Performance Comparison of Palm Tree Coplanar Vivaldi Antenna with Different Patch Structure”, to the Electronics (MDPI).

We express gratitude and give appreciation to the editors and reviewers who are still considering our manuscript in the Electronics (MDPI). We also give greatly gratitude for the positive feedback for the significant improvement to our manuscript.

In this revised manuscript, we include three files (Please see attachment):

  1. Cover Letter
  2. Response to editor and reviewer.
  3. Main file as the final revised manuscript.
  4. Revised manuscript that has been marked up using the “Track
    Changes” function if you are using MS Word, Marked copy of revised document by red letter which indicates the added information.

In this revised, we have tried to do the best revision based on all inputs from reviewers

Once again we appreciate for all positive feedback. We really hope our revised manuscript can be accepted and suitable for publication in the Electronics (MDPI)..

We look forward for further consideration.

Sincerely

Nurhayati

Reviewer 3 Report

Simulation of fifteen antenna of Palm Tree Coplanar Vivaldi in the E-plane was presented, and two types of these antennas were fabricated.  However, there are some questions and comments regarding to the manuscript as follows:

1-    In most of the simulated antennas, the main structure is fixed, and only the outer boundary of the antenna was changed. It is predictable that the antenna characteristics will not change significantly. A reader is not interested in verifying so many antennas and gets confused. Furthermore, there are some more important parameters that can affect the Vivaldi antenna characteristics, but have not been investigated.

2-    What do you mean by the following sentence in the Introduction?

“At low frequencies, Mutual coupling is good at a return loss of less than -20 dB.”

3-    Directivity curves are plotted in Figures 6 and 7. It is not specified that these curves are for a single antenna element, or for the two element array. When discussing MIMO, single-element directivity is expected. So, if these curves are for a single element, why does the directivity decreases, when the frequency increases from 4GHz to 4.5 GHz?

4-    Comment number 3 can also be expressed for the side lobe level shown in Figure 8.

5-    Please explain the reason of beam squint for the antennas in Fig. 9. For a Vivaldi antenna, an end-fire pattern (without beam squint) is expected. Is the simulated pattern is for a single antenna or for the array?

6-    Again in Figure 10, we expect the beam-width for a single antenna element (because of MIMO application). Are the curves shown for the single elements? This question can also be mentioned for Figure 11.

7-    The radiation pattern of RPT-CVA in Figure 11a and 11b is the same as its radiation pattern in Figure 11c and 11d, while they are for two different frequencies (2 and 4 GHz).

8-     Figure 11 is the radiation pattern in the E-plane. You can Also add a new figure for the radiation pattern in the H-plane.

9-    The paragraph after Figure 14 describes the application of a Network analyzer. this is not necessary for a research paper.

10-   The paper should be revised for some English writing and grammar errors.

Author Response

(The authors gave the same response as above.)

Round 2

Reviewer 1 Report

The authors carefully address the comments, thus, the paper is recommended for publication in present form.

Author Response

Thank you for your suggestions to enhance this manuscript, and thank you for taking the time to consider our manuscript to this Journal.
Once again thank you very much . 

Reviewer 3 Report

Some of the previous comments have been done, and the manuscript is revised. In addition, there are still the following questions and comments:

1-      According to the Comment #1 (in the previous comments), I expected that the manuscript to be shorter. A reader is not interested in verifying so many antennas and gets confused.

2-    The main ambiguity of the manuscript is whether you want to study a MIMO antenna or an array antenna? If your antenna is a MIMO antenna, the results must be for a single antenna element. In addition, if your research is an antenna array, what does S21 mean? (An antenna array has one input.)

3-      In Figures 6 and 7, you have added the directivity of the RPT-CVA. Why does its directivity decreases, when the frequency increases from 4GHz to 4.5 GHz? It should be an increasing curve.

4-      I think the authors didn’t’ understand the comment #7 (in the previous comments). The radiation pattern of RPT-CVA appears to be the same in Figures 11 a, b, c, and d. Figures 11 a and b are for 2 GHz, while Figures 11 c and d are for 4 GHz. The radiation pattern of the RPT-CVA in two different frequencies should be different.

5-      There are still some typing and writing errors in the manuscript. For example:

-           “The Constanta of wavy slot based on equation (3) In this case, …”

-          Figure 1. The 1×2 Copplanar Vivaldi Antenna of (a) Regular Regular Palm Tree-Coplanar”

-          The last paragraph in the Introduction.

Author Response

Response to Reviewer 3 Comments(round 2)

Some of the previous comments have been done, and the manuscript is revised. In addition, there are still the following questions and comments:

Point 1: Comments to the Author:

  1. According to the Comment #1 (in the previous comments), I expected that the manuscript to be shorter. A reader is not interested in verifying so many antennas and gets confused.

Author's response:

Thank you so much for your suggestions and feedback. However, we regret since we believe that by giving a variety of designs, readers will be able to provide valuable feedback. Although we exhibit several models, in figures 2, 3, and 4, we only compare three antenna models for each figure, and each figure is compared to the Regular (conventional) model to assess which return loss and isolation performance is superior. In Figure 5, we compare five models for each image. The scattering parameter graph comparison is not too extensive in one graph, and this scattering parameter offers information about the performance of minimizing mutual coupling at low frequencies. Figures 4 and 5 are directivity antenna comparison images. Figure 4 compares the directivity of the 6 antenna models and Figure 5 compares the 5 antenna models but the directivity comparison between the models in each image is still clear. This directivity comparison can be a consideration for choosing which antenna design has the best directivity at the desired frequency. Similarly, Figures 8, 9, 10 can provide information to the antenna reader which model has the SLL, Beamsquint and Beamwidth performance as expected. Once again, we apologize if there are any shortcomings in writing this article

Point 2: Comments to the Author:

  1. The main ambiguity of the manuscript is whether you want to study a MIMO antenna or an array antenna? If your antenna is a MIMO antenna, the results must be for a single antenna element. In addition, if your research is an antenna array, what does S21 mean? (An antenna array has one input.)

Author's response:

Thank you for your suggestions.

We simulate 1x2 MIMO antennas for all models in this article. In addition, we introduced a single element to the RPT model to compare the directivity, SLL, beamsquint, and beamwidth performance of single and 1x2 MIMO antennas. We apologize for previously using the term "array antenna" in our article. However, we have altered the term array to MIMO antenna.  Thanks for the justification suggestion

Point 3: Comments to the Author:

  1. In Figures 6 and 7, you have added the directivity of the RPT-CVA. Why does its directivity decreases, when the frequency increases from 4GHz to 4.5 GHz? It should be an increasing curve.

Author's response:

Thank you. We have added statement in line 296-304: “However, in a wideband antenna, the wider the antenna bandwidth, the greater distance between elements relative to their wavelength (especially at high-end frequencies), and this cause a grating lobe which will reduce the directivity of the antenna. The grating lobe is a side lobe that is enlarged and resembles the main lobe, this thing caused by the effect of changing the distance between the antenna elements further apart. In this case the 1×2 MIMO has spacing between element is 275mm and it means that at a frequency of 0.5 GHz, the distance between elements is 0.458λ while at a frequency of 4.5 GHz the distance between elements is 4.125λ. The distance between elements rises at 4.5 GHz, causing the grating lobe and antenna directivity to diminish.

Point 4: Comments to the Author:

  1. I think the authors didn’t’ understand the comment #7 (in the previous comments). The radiation pattern of RPT-CVA appears to be the same in Figures 11 a, b, c, and d. Figures 11 a and b are for 2 GHz, while Figures 11 c and d are for 4 GHz. The radiation pattern of the RPT-CVA in two different frequencies should be different.

Author's response:

Thank you for the input and apologize if there is a misperception. We have written in a special article for performance comparison of radiation patterns in the 2 GHz frequency in line 401-407:

“At a frequency of 2 GHz, the main lobe RPT is 8.89dBi, the side lobe level (SLL) is -0.7 dB, the Angular width (3dB) is 15.20 and the main lobe direction is 00. The FCPT structure results of 11.6 dBi main lobe, -4.7 dB sidelobe level, 14.10 angular widths, and 00 main lobe direction. While the HWSPT structure generates a major lobe of 11.7 dBi, the main lobe with an angular width of 14.60, and the main lobe direction of 00, as well as a sidelobe level of -4.6 dB. The antenna performance of the FCPT and HWSPT structures at 2 GHz is superior to RPT in the main lobe, SLL, and Angular width(3dB)”.

Based on these data, it can be inferred that the FCPT and HWSPT to RPT structures have an improvement in directivity performance of 2.71dBi and 2.81dBi, respectively, and a decline in SLL of 4 dB and 3.9 dB at the 2 GHz frequency.

Point 5: Comments to the Author:

  1. There are still some typing and writing errors in the manuscript. For example:

 “The Constanta of wavy slot based on equation (3) In this case, …”

Figure 1. The 1×2 Copplanar Vivaldi Antenna of (a) Regular Regular Palm Tree-Coplanar

The last paragraph in the Introduction.

Author's response:

Thank you for your input to improve this article.

For sentence statements

“The Constanta of wavy slot based on equation (3) In this case, …”

We have revised it to:

“ The Constant of the wavy slot in equation (3) is B1= 2, B2= 1, B3= 1, B4= 5, and B5= 36”

We have also removed the double letters Copplanar and words: “Regular

Figure 1. The 1×2 Copplanar Vivaldi Antenna of (a) Regular Palm Tree-Coplanar

Figure 1. The 1×2 Coplanar Vivaldi Antenna of (a) Regular Palm Tree

We have revised the last paragraph in the introduction to:

This paper is organized as follows: section 2 discusses antenna design, section 3 discusses scattering parameter performance, section 4 discusses directivity performance and side lobe level, section 5 discusses beam squint and beamwidth performance, section 6 discusses measurement results and comparisons with similar antennas, and section 7 concludes.
